# Faraday rotation and transmittance as markers of topological phase transitions in 2D materials

Manuel Calixto[1,2⋆], Alberto Mayorgas[1†], Nicolás A. Cordero[2,3,4‡],
Elvira Romera[2,5∘] and Octavio Castaños[6§]

**1** Department of Applied Mathematics, University of Granada,
Fuentenueva s/n, 18071 Granada, Spain
**2** Institute Carlos I for Theoretical and Computational Physics (iC1),
Fuentenueva s/n, 18071 Granada, Spain
**3** Departamento de Física, Universidad de Burgos,
Plaza Misael Bañuelos s/n, 09001 Burgos, Spain
**4** International Research Center in Critical Raw Materials for Advanced Industrial
Technologies (ICCRAM), Plaza Misael Bañuelos s/n, 09001 Burgos, Spain
**5** Department of Atomic, Molecular and Nuclear Physics, University of Granada,
Fuentenueva s/n, 18071 Granada, Spain
**6** Instituto de Ciencias Nucleares, Universidad Nacional Autonoma de Mexico,
Apdo. Postal 70-543, 04510, CDMX, Mexico

⋆ calixto@ugr.es , † albmayrey97@ugr.es , ‡ ncordero@ubu.es ,
∘ eromera@ugr.es , § ocasta@nucleares.unam.mx

## Abstract

We analyze the magneto-optical conductivity (and related magnitudes like transmittance and Faraday rotation of the irradiated polarized light) of some elemental two-dimensional Dirac materials of group IV (graphene analogues, buckled honeycomb lattices, like silicene, germanene, stannane, etc.), group V (phosphorene), and zincblende heterostructures (like HgTe/CdTe quantum wells) near the Dirac and gamma points, under out-of-plane magnetic and electric fields, to characterize topological-band insulator phase transitions and their critical points. We provide plots of the Faraday angle and transmittance as a function of the polarized light frequency, for different external electric and magnetic fields, chemical potential, HgTe layer thickness and temperature, to tune the material magneto-optical properties. We have shown that absortance/transmittance acquires extremal values at the critical point, where the Faraday angle changes sign, thus providing fine markers of the topological phase transition. In the case of non-topological materials as phosphorene, a minimum of the transmittance is also observed due to the energy gap closing by an external electric field.

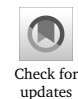

## 1 Introduction

Two-dimensional (2D) materials have been extensively studied in recent years (and are expected to be one of the crucial research topics in future years) especially because of their remarkable electronic and magneto-optical properties which make them hopeful candidates for next generation optoelectronic devices. Graphene is the archetype of a 2D nanomaterial with exceptional high tensile strength, electrical conductivity, transparency, etc. In spite of being the thinnest one, it exhibits a giant Faraday rotation ($\Theta_F \sim 6°$) on polarized light in single- and multilayer arrangements [1–6] with experimental confirmation [7]. Faraday rotation is a fundamental magneto-optical phenomenon used in various optical control, laser technology and magnetic field sensing techniques.

Magneto-optical properties of other buckled honeycomb lattices, like silicene [8], have been studied in [9–11], together with other monolayer transition metal dichalcogenides [12] and anisotropic versions like phosphorene [13]. Magneto-optical measurements also provide signatures of the topological phase transition (TPT; see [14–16] for standard textbooks on the subject) in inverted HgTe/CdTe quantum wells (QW), distinguishing quantum Hall (QH) from quantum spin Hall (QSH) phases [17], where one can tune the band structure by fabricating QWs with different thicknesses $\lambda$. A universal value of the Faraday rotation angle, close to the fine structure constant, has been experimentally observed in thin HgTe QW with critical thickness [18].

To determine experimentally the Faraday rotation effect in Dirac materials it is convenient to consider: (1) A transverse-magnetic-polarized wave incident from the left onto a single

topological insulator sandwiched by dielectric layers, which yields an enhancement of the Faraday rotation with an angle larger than 700 mrad and with a transmission higher than 90% [19]. (2) A graphene sheet sandwiched by one-dimensional topological photonic crystals also an enhancement of the Faraday rotation can be achieved with high transmittance [20]. (3) In thin films of 3D topological insulators [21] or by considering thin films of Floquet topological insulators where giant Faraday and Ker rotations have been observed under the action of a perpendicular magnetic field or in a non-resonant optical field [22]. The inverse Faraday effect (IFE) has been studied in Dirac materials in 2D and 3D, and these studies have concluded that IFE is stronger than in conventional semiconductors. Then the Dirac materials can be potentially useful for the optical control of magnetization in optoelectronic devices [23].

Information theoretic measures also provide signatures of the TPT in silicene [24–28] and HgTe/CdTe QWs [29], as an alternative to the usual topological (Chern) numbers. They also account for semimetalic behavior of phosphorene [30,31] under perpendicular electric fields.

In this paper we perform a comparative study of the magneto-optical properties of several 2D Dirac materials, looking for TPT signatures when the band structure is tuned by applying external fields or by changing the material characteristics. For this purpose, we focus on transmittance and Faraday rotation near the critical point of the topological phase transition for topological materials such as silicene and HgTe quantum wells. We found that, for these materials, transmittance attains an absolute minimum $\mathcal{T}_0$ at the critical TPT point for a certain value $\Omega_0$ of the normal incident polarized light frequency. This minimal behavior does not depend on the chosen values of magnetic field, chemical potential and temperature, although the location of $\Omega_0$ varies with them. An inflection point of the Faraday angle is observed at each peak of the transmittance, coinciding in frequency. As a novel perspective, we study that non-topological materials, such as phosphorene, also exhibit an extremal value of the transmittance when the energy gap is closed by an external electric field.

The organization of the article is as follows. In Sec. 2 we discuss the structure of time independent Bloch Hamiltonians for general two-band 2D-Dirac material models, their Chern numbers and their minimal coupling to an external perpendicular magnetic field. We particularize to graphene analogues (silicene, germanene, etc.) in Sec. 2.1, zincblende heterostructures (HgTe/CdTe quantum wells) in Sec. 2.2 and anisotropic materials like phosphorene in 2.3, calculating their energy spectrum and Hamiltonian eigenstates (Landau levels) and describing their topological phases (when they exist). In Sec. 3 we recall the Kubo-Greenwood formula for the magneto-optical conductivity tensor $\boldsymbol{\sigma}$ of a 2D electron system in a perpendicular magnetic field $B$ and an oscillating electric field of frequency $\Omega$. In particular, we are interested in analyzing the transmittance and Faraday rotation of linearly polarized light of frequency $\Omega$ for normal incidence on the 2D material. Magneto-optical properties of graphene analogues, zincblende heterostructures and phosphorene are analyzed in Sections 3.1, 3.2 and 3.3, respectively. For topological insulator materials, we find that the critical point is generally characterized by a minimum transmittance $\mathcal{T}_0$ at a given light frequency $\Omega_0$, where the Faraday angle changes sign. The effect of anisotropies is also discussed in phosphorene in Section 3.3. Finally, Sec. 4 is devoted to conclusions.

## 2 Some two-band 2D-Dirac material models

The time independent Bloch Hamiltonian of a two-band 2D insulator has the general form

$$H(\boldsymbol{k}) = \epsilon_0(\boldsymbol{k})\tau_0 + \boldsymbol{d}(\boldsymbol{k}) \cdot \boldsymbol{\tau}, \tag{1}$$

where $\boldsymbol{\tau} = (\tau_x, \tau_y, \tau_z)$ is the Pauli matrix vector, $\tau_0$ denotes the $2 \times 2$ identity matrix and $\boldsymbol{d}(\boldsymbol{k})$ parameterizes an effective spin-orbit coupling near the center $\Gamma$ or the Dirac valleys $K$

and $K'$ of the first Brillouin zone (FBZ), with $\boldsymbol{k} = (k_x, k_y)$ the two-dimensional wavevector. The energy of the two bands is $\epsilon_\pm(\boldsymbol{k}) = \epsilon_0(\boldsymbol{k}) \pm |\boldsymbol{d}(\boldsymbol{k})|$.

To distinguish between band insulator and topological insulator phases, one can use the TKNN (Thouless-Kohmoto-Nightingale-Nijs) formula [32] providing the Chern-Pontryagin number (related to the quantum spin Hall conductance and the Berry phase [33])

$$\mathcal{C} = \frac{1}{2\pi} \int \int_{\text{FBZ}} d^2\boldsymbol{k} \left( \frac{\partial \hat{\boldsymbol{d}}(\boldsymbol{k})}{\partial k_x} \times \frac{\partial \hat{\boldsymbol{d}}(\boldsymbol{k})}{\partial k_y} \right) \cdot \hat{\boldsymbol{d}}(\boldsymbol{k}), \tag{2}$$

with $\hat{\boldsymbol{d}} = \boldsymbol{d}/|\boldsymbol{d}|$, which counts the number of times (winding number) the unit vector $\hat{\boldsymbol{d}}(\boldsymbol{k})$ wraps around the unit sphere as $\boldsymbol{k}$ wraps around the entire FBZ. The Chern number $\mathcal{C}$ usually depends on the sign of some material and (external) control parameters in the Hamiltonian $H$ (see later for some examples), taking different values in different phases. We shall see that magneto-optical conductivity measures also capture the topological phase transition.

We shall consider the interaction with a perpendicular magnetic field $\boldsymbol{B} = (0, 0, B)$. Promoting the wavevector $\boldsymbol{k}$ to the momentum operator $\boldsymbol{k} \to \boldsymbol{p}/\hbar = -i\boldsymbol{\nabla}$, this interaction is introduced through the usual minimal coupling, $\boldsymbol{p} \to \boldsymbol{P} = \boldsymbol{p} + e\boldsymbol{A}$ with $\boldsymbol{A} = (A_x, A_y) = (-By, 0)$ the electromagnetic potential (in the Landau gauge) and $e$ the elementary charge (in absolute value). After Peierls' substitution, which results in

$$k_x \to P_x/\hbar = \frac{a^\dagger + a}{\sqrt{2}\ell_B}, \qquad k_y \to P_y/\hbar = \frac{a^\dagger - a}{i\sqrt{2}\ell_B}, \tag{3}$$

the Hamiltonian (1) can be eventually written in terms of creation $a^\dagger$ and annihilation

$$a = \frac{\ell_B}{\sqrt{2}\hbar}(P_x - iP_y) = \frac{-1}{\sqrt{2}\ell_B}(y - y_0 + i\ell_B^2 p_y/\hbar), \tag{4}$$

operators, where $\ell_B = \sqrt{\hbar/(eB)}$ is the magnetic length and $y_0 = \ell_B^2 k_x$ is the coordinate of the conserved center of the cyclotron orbit.

Let us review some relevant physical examples.

## 2.1 Graphene analogues: Silicene, germanene, etc

Silicene, germanene, and other transition metal dichalcogenides (of the Xene type) exhibit an intrinsic non-zero spin-orbit coupling $H_{\text{so}} = -\frac{1}{2}s\xi\Delta_{\text{so}}\tau_z$ ($s = \pm 1$ is the spin of the electron and $\xi = \pm 1$ refer to the Dirac valleys $K$ and $K'$) due to second neighbors hopping terms in the tight binding model [34]. Spin-orbit interaction $H_{\text{so}}$ combined with and external perpendicular electric field coupling $H_{\Delta_z} = \frac{1}{2}\Delta_z\tau_z$, gives $\boldsymbol{d}(\boldsymbol{k}) = (\nu\hbar\xi k_x, \nu\hbar k_y, \Delta_{s\xi})$, where $\Delta_{s\xi} = (\Delta_z - s\xi\Delta_{\text{so}})/2$ results in a tunable (Dirac mass) gap (see e.g. [35–38]). In Table 1 we show a comparative of spin-orbit coupling and Fermi velocity values for several 2D materials.

The Chern number (2) turns out to be

$$\mathcal{C}_{s\xi} = \xi\,\text{sign}(\Delta_{s\xi}), \tag{5}$$

where we have integrated on the whole plane, as corresponds to the FBZ in the continuum limit (zero lattice constant). Therefore, the topological phase is determined by the sign of the Dirac mass at each valley $\xi$. More precisely, there is a TPT from a topological insulator (TI, $|\Delta_z| < \Delta_{\text{so}}$) to a band insulator (BI, $|\Delta_z| > \Delta_{\text{so}}$), at a charge neutrality point (CNP) $\Delta_z^{(0)} = s\xi\Delta_{\text{so}}$, where there is a gap cancellation between the perpendicular electric field and the spin-orbit coupling.

Table 1: Approximate values of model parameters $\Delta_{so}$ (spin-orbit coupling), $l$ (inter-lattice distance) and $v$ (Fermi velocity) for two dimensional Si, Ge, Sn and Pb sheets. These data have been obtained from first-principles computations in [38] ($\Delta_{so}$ and $l$) and [39,40] ($v$).

|     | $\Delta_{so}$ (meV) | $l$ (Å) | $v$ ($10^5$m/s) |
|-----|-----|-----|-----|
| Si  | 4.2   | 0.22 | 4.2 |
| Ge  | 11.8  | 0.34 | 8.8 |
| Sn  | 36.0  | 0.42 | 9.7 |
| Pb  | 207.3 | 0.44 | –   |

Using the general prescription (3), the minimal coupling with a perpendicular magnetic field $B$ then results in a different Hamiltonian $H_\xi$ for each valley $\xi = \pm 1$

$$H_1 = \begin{pmatrix} \Delta_{s,1} & \hbar\omega a \\ \hbar\omega a^\dagger & -\Delta_{s,1} \end{pmatrix}, \qquad H_{-1} = \begin{pmatrix} \Delta_{s,-1} & -\hbar\omega a^\dagger \\ -\hbar\omega a & -\Delta_{s,-1} \end{pmatrix}, \tag{6}$$

where $\omega = \sqrt{2}v/\ell_B$ denotes the cyclotron frequency. The eigenvalues of both Hamiltonians are simply:

$$E_n^{s\xi} = \begin{cases} \text{sgn}(n)\sqrt{|n|\hbar^2\omega^2 + \Delta_{s\xi}^2}, & n \neq 0, \\ -\xi\Delta_{s\xi}, & n = 0, \end{cases} \tag{7}$$

and the corresponding eigenstates are written in terms of Fock states $||n\rangle$, for Landau level (LL) index $n = 0, \pm 1, \pm 2, \dots$ [valence ($-$) and conduction ($+$) states], as spinors

$$|\boldsymbol{n}\rangle_{s\xi} = \begin{pmatrix} A_n^{s\xi}\left||n| - \frac{\xi+1}{2}\right\rangle \\ B_n^{s\xi}\left||n| + \frac{\xi-1}{2}\right\rangle \end{pmatrix}, \tag{8}$$

with coefficients (see [9,41–43] for similar results)

$$A_n^{s\xi} = \begin{cases} \frac{\text{sgn}(n)}{\sqrt{2}}\sqrt{1 + \text{sgn}(n)\cos\theta_n^{s\xi}}, & n \neq 0, \\ (1-\xi)/2, & n = 0, \end{cases}$$

$$\tag{9}$$

$$B_n^{s\xi} = \begin{cases} \frac{\xi}{\sqrt{2}}\sqrt{1 - \text{sgn}(n)\cos\theta_n^{s\xi}}, & n \neq 0, \\ (1+\xi)/2, & n = 0, \end{cases}$$

where $\theta_n^{s\xi} = \arctan\left(\hbar\omega\sqrt{|n|}/\Delta_{s\xi}\right)$, that is, $\cos\theta_n^{s\xi} = \Delta_{s\xi}/|E_n^{s\xi}|$. Note that $A_n^{s\xi}$ and $B_n^{s\xi}$ can eventually be written as $\cos(\theta_n^{s\xi}/2)$ or $\sin(\theta_n^{s\xi}/2)$, depending on $\text{sgn}(n)$.

In Figure 1 we plot the low energy spectra of silicene, given by (7), as a function of the external electric field $\Delta_z$, together with the charge neutrality (critical) points $\Delta_z^{(0)} = \pm|\Delta_{so}|$ (marked by vertical dashed lines) at which the TPT takes place.

## 2.2 HgTe/CdTe quantum wells

In [44–47] it was shown that quantum spin Hall effect can be realized in mercury telluride-cadmium telluride semiconductor quantum wells. Similar effects were also predicted in Type-II semiconductor quantum wells made from InAs/GaSb/AlSb [48]. The surface states in these 3D topological insulators can be described by a 2D modified effective Dirac Hamiltonian

$$H = \begin{pmatrix} H_+ & 0 \\ 0 & H_- \end{pmatrix}, \qquad H_s(\boldsymbol{k}) = \epsilon_0(\boldsymbol{k})\tau_0 + \boldsymbol{d}_s(\boldsymbol{k})\cdot\boldsymbol{\tau}, \tag{10}$$

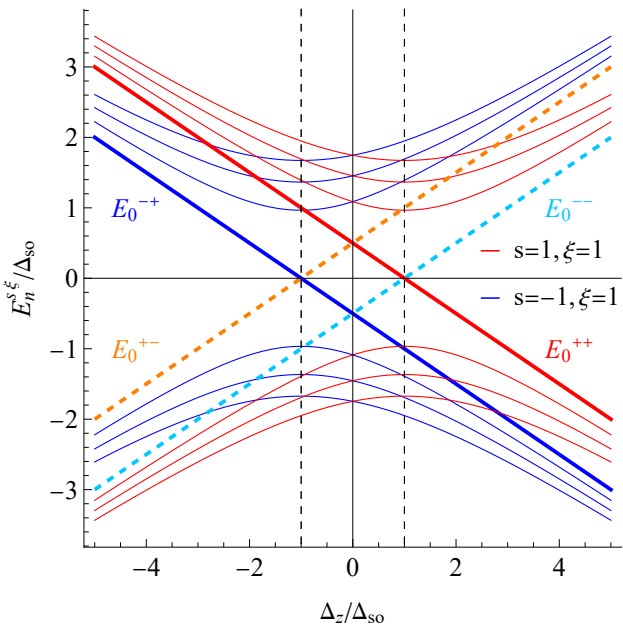

Figure 1: Low energy spectra of silicene as a function of the external electric potential $\Delta_z$ (in $\Delta_{so}$ units) for $B = 0.05$ T. Landau levels $n = \pm1, \pm2$ and $\pm3$ [valence $(-)$ and conduction $(+)$], at valley $\xi = 1$, are represented by thin solid lines, blue for $s = -1$ and red for $s = 1$ (for the other valley we simply have $E_n^{s,-\xi} = E_n^{-s,\xi}$). The edge states $n = 0$ are represented by thick lines at both valleys: solid at $\xi = 1$ and dashed at $\xi = -1$. Vertical dashed gray lines indicate the charge neutrality points separating band insulator ($|\Delta_z| > \Delta_{so}$) from topological insulator ($|\Delta_z| < \Delta_{so}$) phases.

where $s = \pm1$ is the spin and $H_-(\boldsymbol{k}) = H_+^*(-\boldsymbol{k})$ (temporarily reversed). The expansion of $H_s(\boldsymbol{k})$ about the center $\Gamma$ of the first Brillouin zone gives [45]

$$\epsilon_0(\boldsymbol{k}) = \gamma - \delta \boldsymbol{k}^2, \quad \boldsymbol{d}_s(\boldsymbol{k}) = (\alpha s k_x, \alpha k_y, \mu - \beta \boldsymbol{k}^2), \tag{11}$$

where $\alpha, \beta, \gamma, \delta$ and $\mu$ are expansion parameters that depend on the heterostructure (the HgTe layer thickness $\lambda$). The most important one is the mass or gap parameter $\mu$, which changes sign at a critical HgTe layer thickness $\lambda_c$ when going from the normal ($\lambda < \lambda_c$ or $\mu/\beta < 0$) to the inverted ($\lambda > \lambda_c$ or $\mu/\beta > 0$) regime [49]. Typical values of these parameters for different HgTe layer thickness (below and above $\lambda_c$) can be found in [49] and in Table 2 ($\gamma$ can be neglected).

The energy of the two bands is

$$\epsilon_\pm(\boldsymbol{k}) = \epsilon_0(\boldsymbol{k}) \pm \sqrt{\alpha^2 \boldsymbol{k}^2 + (\mu - \beta \boldsymbol{k}^2)^2}. \tag{12}$$

Table 2: Material parameters for HgTe/CdTe quantum wells with different HgTe layer thicknesses $\lambda$ [49].

| $\lambda$(nm) | $\alpha$(meV·nm) | $\beta$(meV·nm²) | $\delta$(meV·nm²) | $\mu$(meV) |
|---|---|---|---|---|
| 5.5 | 387 | -480 | -306 | 9 |
| 6.1 | 378 | -553 | -378 | -0.15 |
| 7.0 | 365 | -686 | -512 | -10 |

The TKNN formula (2) for $\boldsymbol{d}_s(\boldsymbol{k})$ provides the Chern number

$$\mathcal{C}_s = s[\text{sign}(\mu) + \text{sign}(\beta)], \tag{13}$$

where we have integrated on the whole plane, as corresponds to the continuum limit. According to Table 2, $\beta$ does not change sing and, therefore, the topological phase transition occurs when $\mu$ changes sign, as already mentioned. In reference [49], the normal and inverted regimes are equivalently given by the sign of $\mu/\beta$.

Using again the general prescription (3), the minimal coupling with a perpendicular magnetic field $B$ now results in

$$H_+ = \begin{pmatrix} \gamma + \mu - \frac{(\delta+\beta)(2N+1)}{\ell_B^2} & \frac{\sqrt{2}\alpha}{\ell_B}a \\ \frac{\sqrt{2}\alpha}{\ell_B}a^\dagger & \gamma - \mu - \frac{(\delta-\beta)(2N+1)}{\ell_B^2} \end{pmatrix},$$

$$H_- = \begin{pmatrix} \gamma + \mu - \frac{(\delta+\beta)(2N+1)}{\ell_B^2} & -\frac{\sqrt{2}\alpha}{\ell_B}a^\dagger \\ -\frac{\sqrt{2}\alpha}{\ell_B}a & \gamma - \mu - \frac{(\delta-\beta)(2N+1)}{\ell_B^2} \end{pmatrix}, \tag{14}$$

with $N = a^\dagger a$. A Zeeman term contribution

$$H_s^Z = -\frac{s}{2}B\mu_{\text{B}}\left(g_{\text{e}}\frac{\tau_0 + \tau_z}{2} + g_{\text{h}}\frac{\tau_0 - \tau_z}{2}\right), \tag{15}$$

can also be added to the Hamiltonian, with $\mu_{\text{B}} \simeq 0.058$ meV/T the Bohr magneton and $g_{\text{e,h}}$ the effective (out-of-plane) $g$-factors for electrons and holes (conduction and valence bands).

Using (Fock state) eigenvectors $||n\rangle$ of the (Landau level) number operator $N = a^\dagger a$, one can analytically obtain the eigenspectrum

$$E_n^s = \gamma - \frac{2\delta|n|-s\beta}{\ell_B^2} - s\frac{g_{\text{e}}+g_{\text{h}}}{4}B\mu_{\text{B}} + \text{sgn}(n)\sqrt{\frac{2\alpha^2|n|}{\ell_B^2} + \left(\mu - \frac{2\beta|n|-s\delta}{\ell_B^2} - s\frac{g_{\text{e}}-g_{\text{h}}}{4}B\mu_{\text{B}}\right)^2}, \tag{16}$$

for LL index $n = \pm 1, \pm 2, \pm 3, \dots$ [valence $(-)$ and conduction $(+)$] , and

$$E_0^s = \gamma - s\mu - \frac{\delta - s\beta}{\ell_B^2} - B\mu_{\text{B}}\left(\frac{s+1}{4}g_{\text{h}} + \frac{s-1}{4}g_{\text{e}}\right), \tag{17}$$

for the edge states $n = 0$, $s = \pm 1$. These eigenvalues coincide with those in [17, 50, 51] for the identification $s = \{-1, 1\} = \{\uparrow, \downarrow\}$.

The corresponding eigenvectors are

$$|\boldsymbol{n}\rangle_s = \begin{pmatrix} A_n^s \left||n| - \frac{s+1}{2}\right\rangle \\ B_n^s \left||n| + \frac{s-1}{2}\right\rangle \end{pmatrix}, \tag{18}$$

with coefficients

$$A_n^s = \begin{cases} \frac{\text{sgn}(n)}{\sqrt{2}}\sqrt{1 + \text{sgn}(n)\cos\vartheta_n^s}, & n \neq 0, \\ (1-s)/2, & n = 0, \end{cases} \tag{19}$$

$$B_n^s = \begin{cases} \frac{s}{\sqrt{2}}\sqrt{1 - \text{sgn}(n)\cos\vartheta_n^s}, & n \neq 0, \\ (1+s)/2, & n = 0, \end{cases}$$

where

$$\vartheta_n^s = \arctan\left(\frac{\sqrt{2|n|}\,\alpha/\ell_B}{\mu - \frac{2\beta|n|-s\delta}{\ell_B^2} - s\frac{g_e-g_h}{4}B\mu_B}\right). \tag{20}$$

As for the graphene analogues in (9), the coefficients $A_n^s$ and $B_n^s$ can eventually be written as sine and cosine of half angle, depending on $\mathrm{sgn}(n)$.

According to (17), the band inversion for edge states occurs when

$$E_0^+ = E_0^- \Rightarrow B_{\mathrm{inv}} = \frac{\mu}{e\beta/\hbar - \mu_B(g_e + g_h)/4}, \tag{21}$$

which gives the critical magnetic field $B_c$ which separates the QSH and QH regimes [51]. For example, for the material parameters in Table 2 corresponding to a QW thickness $\lambda = 7.0$ nm and $g$-factors $g_e = 22.7, g_h = -1.21$, one obtains $B_{\mathrm{inv}} \simeq 7.4$ T. See also Figure 2 for a graphical representation of this band inversion.

From now on we shall discard Zeeman coupling for the sake of convenience since our main conclusions remain qualitatively equivalent. We address the interested reader to Appendix C where we reproduce some results of Reference [17] for non-zero Zeeman coupling and contrast with the zero Zeeman coupling case.

We shall use a linear fit

$$\begin{aligned}
\mu(\lambda) &= 77.31 - 12.53\lambda, \\
\alpha(\lambda) &= 467.49 - 14.65\lambda, \\
\beta(\lambda) &= 283.58 - 138.16\lambda, \\
\delta(\lambda) &= 458.46 - 138.25\lambda,
\end{aligned} \tag{22}$$

of the material parameters in Table 2 as a function of the HgTe layer thickness $\lambda$ (dimensionless units and $\lambda$ in nm units). In all cases the coefficient of determination is $R^2 > 0.99$. Looking at $\mu(\lambda)$ in (22), we can obtain an estimation of the critical HgTe thickness at which the topological phase transition occurs as

$$\mu = 0 \Rightarrow \lambda_c = 6.17 \text{ nm}. \tag{23}$$

In Figure 2 we plot the low energy spectra given by (16) and (17) as a function of the HgTe layer thickness $\lambda$, where we have extrapolated the linear fit (22) to the interval $[4\,\mathrm{nm}, 8\,\mathrm{nm}]$. When neglecting Zeeman coupling, the band inversion for edge states (21) occurs for $B = \hbar\mu/(e\beta)$ which, using the linear fit (22), provides a relation

$$\lambda_{\mathrm{inv}}(B) = \frac{368.31 - 2.05B}{59.7 - B}, \tag{24}$$

between the applied magnetic field $B$ (in Tesla) and the HgTe layer thickness $\lambda_{\mathrm{inv}}(B)$ (in nanometers) at which the band inversion $E_0^+ = E_0^-$ takes place. Note that $\lambda_{\mathrm{inv}}(B) \simeq \lambda_c = 6.17$ nm for low $B \ll 1$ T, and that $E_0^+ = E_0^- \simeq 0$ meV at this point as shows Figure 2.

## 2.3 Phosphorene as an anisotropic material

The physics of phosphorene has been extensively studied [53–67]. There are several approaches to the low energy Hamiltonian of phosphorene in the literature. Rudenko et al. [68] and Ezawa [69] propose a four-band and five-neighbors tight-binding model later simplified to two-bands [69]. Several approximations of this two-band model have been used in [13,70–72]. We shall choose for our study the Hamiltonian

$$H = \begin{pmatrix} E_c + \alpha_x k_x^2 + \alpha_y k_y^2 & \gamma k_x \\ \gamma k_x & E_v - \beta_x k_x^2 - \beta_y k_y^2 \end{pmatrix}, \tag{25}$$

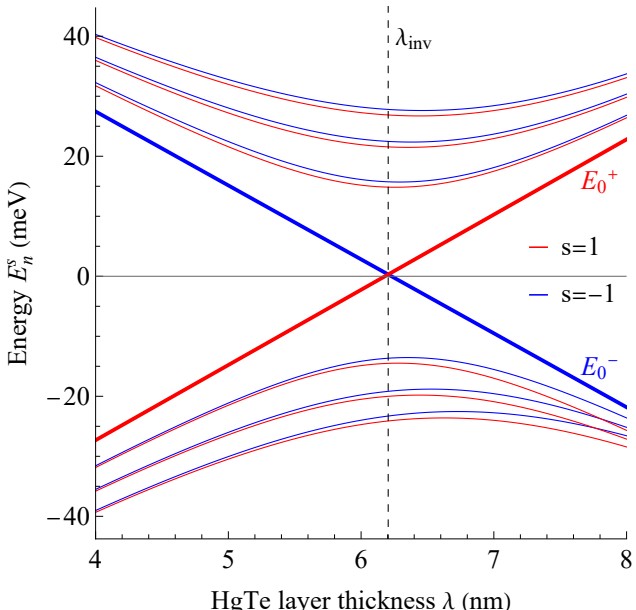

Figure 2: Low-energy spectra $E_n^s$ of a HgTe/CdTe quantum well as a function of the HgTe layer thickness $\lambda$ for $B = 0.5$ T. Landau levels $n = \pm 1, \pm 2, \pm 3$ [valence $(-)$ and conduction $(+)$] are represented by thin solid lines, blue for spin $s = -1$ and red for $s = 1$. Edge states $(n = 0)$ are represented by thick lines. A vertical dashed black line indicates the HgTe thickness $\lambda_{\text{inv}}(0.5) = 6.20$ nm $\simeq \lambda_c$ where the band inversion for edge states occurs for $B = 0.5$ T according to (24).

proposed by Zhou and collaborators [13]. This corresponds to a Bloch Hamiltonian (1) with

$$\epsilon_0(\boldsymbol{k}) = \frac{E_c + E_v + (\alpha_x - \beta_x)k_x^2 + (\alpha_y - \beta_y)k_y^2}{2}, \tag{26}$$

$$\boldsymbol{d}(\boldsymbol{k}) = \left( \gamma k_x, 0, \frac{E_c - E_v + (\alpha_x + \beta_x)k_x^2 + (\alpha_y + \beta_y)k_y^2}{2} \right).$$

The Hamiltonian (25) provides a trivial Chern number (2), even in the presence of a tunable perpendicular constant electric field (see below), which means that monolayer phosphorene does not have a topological phase *per se*. It has been shown that topological transitions can be induced in phosphorene when rapidly driven by in-plane time-periodic laser fields [73]; these are called in general "Floquet topological insulators" (see e.g. [74–76]), but we shall not consider this possibility here. Although phosphorene is not a topological material, we will see in Sec. 3.3 that the critical magneto-optical properties (e.g., minimum transmittance) observed for silicene and HgTe QWs are still valid in phosphorene when closing the energy gap through an external electric field. Another possibility to modify the energy gap could be by applying strain [60, 70] (see later in Sec. 3.3).

The material parameters of phosphorene can be written in terms of conduction (c) and valence (v) effective masses as (see [13] for more information)

$$\alpha_{x,y} = \frac{\hbar^2}{2m_{cx,cy}}, \quad \beta_{x,y} = \frac{\hbar^2}{2m_{vx,vy}}, \tag{27}$$

with

$$\begin{aligned} m_{cx} &= 0.793m_e, \quad m_{cy} = 0.848m_e, \\ m_{vx} &= 1.363m_e, \quad m_{vy} = 1.142m_e, \end{aligned} \tag{28}$$

and $m_e$ is the free electron mass. Conduction and valence band edge energies are $E_c = 0.34$ eV and $E_v = -1.18$ eV, so that the energy gap is $E_g = E_c - E_v = 1.52$ eV. The interband coupling parameter is $\gamma = -0.523$ eV·nm.

When coupling to an external perpendicular magnetic field, the anisotropic character of phosphorene slightly modifies Peierls' substitution (3), which now adopts the following form

$$k_x \rightarrow \frac{P_x}{\hbar} = \frac{a^\dagger + a}{\sqrt{2}\alpha_{yx}\ell_B}, \quad k_y \rightarrow \frac{P_y}{\hbar} = \frac{\alpha_{yx}(a^\dagger - a)}{i\sqrt{2}\ell_B}, \tag{29}$$

with $\alpha_{yx} = \left(\frac{m_{cy}}{m_{cx}}\right)^{1/4}$. Therefore, applying this prescription to (25), the final Hamiltonian can be written as

$$H = \hbar\omega_\gamma(a + a^\dagger)\tau_x + \left[E_c + \hbar\omega_c(a^\dagger a + 1/2)\right]\frac{\tau_0 + \tau_z}{2} \tag{30}$$
$$+ \left[E_v - \hbar\omega_v(a^\dagger a + 1/2) - \hbar\omega'(a^2 + a^{\dagger 2})\right]\frac{\tau_0 - \tau_z}{2},$$

in terms of the annihilation (and creation $a^\dagger$) operator

$$a = \sqrt{\frac{m_{cy}\omega_c}{2\hbar}}\left(y - y_0 + i\frac{\hat{p}_y}{m_{cy}\omega_c}\right), \tag{31}$$

in analogy to (4), where some effective frequencies have been defined as

$$\begin{aligned}
\omega_c &= \frac{eB}{\sqrt{m_{cx}m_{cy}}}, & \omega_\gamma &= \frac{\gamma}{\sqrt{2}\hbar\alpha_{yx}\ell_B}, \\
\omega_v &= (r_x + r_y)\omega_c, & \omega' &= (r_x - r_y)\omega_c/2,
\end{aligned} \tag{32}$$

with

$$r_x = \frac{m_{cx}}{2m_{vx}}, \qquad r_y = \frac{m_{cy}}{2m_{vy}}. $$

As we did for silicene, we shall also consider here the application of a perpendicular electric field to the phosphorene sheet in the usual form [77] $\hat{H}_\Delta = \Delta_z \tau_z$, with $\Delta_z$ the on-site electric potential. Unlike for silicene and HgTe QWs, the diagonalization of the phosphorene Hamiltonian (30) has to be done numerically [30].

Note that the Hamiltonian (30) preserves the parity $\pi(n,s) = e^{i\pi n_s}$ of the state $|n\rangle_s$, with $n_s = n + (s+1)/2$ (see e.g. [30]). This means that the matrix elements $_s\langle n|H|n'\rangle_{s'} \propto \delta_{\pi(n,s),\pi(n',s')}$ are zero between states of different parity. Therefore, this parity symmetry helps in the diagonalization process and any (non-degenerate) eigenstate of $H$ has a definite parity. The Hamiltonian eigenstates can now be written as

$$|\psi_l\rangle = \sum_{n,s} c_{n,s}^{(l)}|n\rangle_s, \tag{33}$$

where $l \in \mathbb{Z}$ denotes the LL index ($l > 0$ for conduction and $l \le 0$ for valence band). The sum $\sum_{n,s}$ is constrained to $\pi(n,s) = \pm 1$, depending on the even (+) and odd (−) parity of $k$. The coefficients $c_{n,s}^{(k)}$ are obtained by numerical diagonalization of the Hamiltonian matrix, which is truncated to $n \le N$, with $N$ large enough to achieve convergent results for given values of the magnetic and electric fields. In particular, we have used Fock states with $N \le 1000$ to achieve convergence (with error tolerance $\le 10^{-15}$ eV) for $B = 0.5$ T in the six first Hamiltonian eigenvalues in the range $-1.55 \le \Delta_z \le -1.49$ eV. The resulting spectrum, as a function of the electric field potential $\Delta_z$, can be seen in Figure 3 for a magnetic field of $B = 0.5$ T (higher magnetic fields need less Fock states to achieve convergence). The vertical dashed

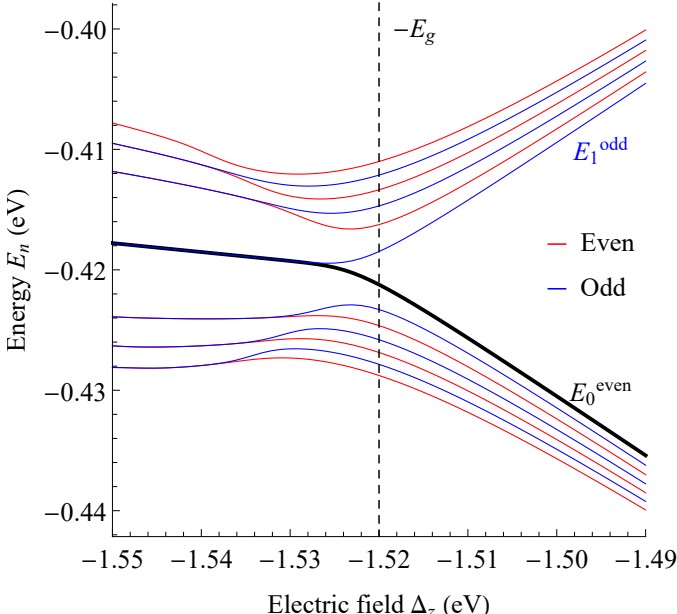

Figure 3: Low energy spectra $E_l$ of phosphorene as function of the electric field potential $\Delta_z$ for thirteen Hamiltonian eigenstates $l = -6, \ldots, 0, \ldots, 6$ and a magnetic field $B = 0.5$ T. Valence and conduction band states of even (odd) parity $l = \pm 2, \pm 4, \pm 6$ ($n = \pm 1, \pm 3, \pm 5$) are represented in red (blue) color. The edge state $E_0^{\text{even}}$ is represented by a thick black line. The vertical dashed black line is the point $\Delta_z = -E_g = -1.520$ eV at which the electric potential equals the energy gap of phosphorene.

line gives the point $\Delta_z = -1.520$ eV at which the electric potential equals minus the energy gap $E_g = E_c - E_v = 1.52$ eV of phosphorene. This is not really a critical point in the same sense as $\Delta_z^{(0)} = \Delta_{so} = 4.2$ meV for silicene and $\lambda_c = 6.17$ nm for HgTe QWs, since phosphorene as such (as already said) does not display a topological phase. However, we will see in Section 3.3 that the phosphorene transmittance still presents a minimum at $\Delta_z^{(0)} = -1.523$ eV, which closes the energy gap $E_g = 1.52$ eV at low magnetic fields.

It is also interesting to note that the LLs of phosphorene are degenerated in pairs for an electric potential below $\Delta_z \simeq -1.53$ eV. Namely, we obtain numerically that $|E_l^{\text{even}} - E_{l+1}^{\text{odd}}| \le 10^{-4}$ eV for all $\Delta_z < -1.53$ eV and $l = -6, -4, -2, 0, 2, 4$ as it shows the left hand side of Figure 3. This energy degeneracy will influence the conductivity as well.

## 3  Magneto-optical conductivity

The magneto-optical conductivity tensor $\boldsymbol{\sigma}$ of a 2D electron system in a perpendicular magnetic field $B$ and an oscillating electric field of frequency $\Omega$, can be obtained from Kubo-Greenwood formula [32,78,79] in the Landau-level representation:

$$\sigma_{ij}(\Omega, B) = \frac{i\hbar}{2\pi\ell_B^2} \sum_{n,m} \frac{f_m - f_n}{E_n - E_m} \frac{\langle \boldsymbol{m}|j_i|\boldsymbol{n}\rangle \langle \boldsymbol{n}|j_j|\boldsymbol{m} >}{\hbar\Omega + E_m - E_n + i\eta}, \tag{34}$$

where

$$\boldsymbol{j} = \frac{ie}{\hbar}[H, \boldsymbol{r}] = \frac{e}{\hbar}\nabla_k H, \tag{35}$$

is the current operator, with $\boldsymbol{r} = (x, y)$ and $\nabla_{\boldsymbol{k}} = (\partial_{k_x}, \partial_{k_y})$ [the minimal coupling prescription (3) is understood under external electromagnetic fields], and $f_n = 1/(1 + \exp[(E_n - \mu_F)/(k_B T)])$ is the Fermi distribution function at temperature $T$ and chemical potential $\mu_F$. In the zero temperature limit, the Fermi function $f_n$ is replaced by the Heaviside step function $\Theta(\mu_F - E_n)$, which enforces the Pauli exclusion principle for optical transitions (they are allowed between occupied and unoccupied states). The parameter $\eta$ is a small residual scattering rate of charge carriers and, although the exact shape of $\sigma_{ij}$ would depend on the details of the scattering mechanisms, using a constant $\eta$ gives a good, qualitative description of the essential mechanisms relevant for magneto-optical experiments. In $\sum_n$ of eq. (34) it is also implicit the sum over spin $s$ and valley $\xi$, besides the LL index $n$ (for graphene, there is a twofold spin and valley degeneracy, so that the extra sum just contributes with a degeneracy factor $g = 4$). We shall measure $\sigma_{ij}$ in units of the conductance quantum $\sigma_0 = e^2/h = 38.8 \ \mu\text{S}$ [78] and renormalize the currents as $\bar{j} = j/(e/\hbar) = \nabla_{\boldsymbol{k}} H$, so that

$$\frac{\sigma_{ij}(\Omega, B)}{\sigma_0} = \frac{\mathrm{i}}{\ell_B^2} \sum_{n,m} \frac{f_m - f_n}{E_n - E_m} \frac{\langle m|\bar{j}_i|n\rangle \langle n|\bar{j}_j|m\rangle}{\hbar\Omega + E_m - E_n + i\eta} \, . \tag{36}$$

We shall analyze the transmittance and Faraday rotation of linearly polarized light of frequency $\Omega$ for normal incidence on the 2D material, where the electric fields of incident ($\boldsymbol{E}^i$) and transmitted ($\boldsymbol{E}^t$) waves are related through the conductivity tensor $\boldsymbol{\sigma}$ by the formula [80–82]

$$\boldsymbol{E}^t = \left(I + \tfrac{1}{2} Z_0 \boldsymbol{\sigma}\right)^{-1} \cdot \boldsymbol{E}^i \, , \tag{37}$$

where $Z_0 = 2\alpha/\sigma_0$ is the vacuum impedance ($\alpha = 1/137$ is the fine-structure constant) and $I$ denotes the $2 \times 2$ identity matrix. We also assume that the incident field is linearly polarized in the $x$ axis, that is $\boldsymbol{E}^i = (E_x^i, 0)$. From here, the transmittance $\mathcal{T}$ and the Faraday rotation angle $\Theta_F$ (in degrees) are [2, 82]

$$\mathcal{T} = \frac{1}{2}(|t_+|^2 + |t_-|^2) \simeq 1 - Z_0 \text{Re}(\sigma_{xx}), \tag{38}$$

$$\Theta_F = \frac{1}{2}(\arg(t_+) + \arg(t_-)) \simeq \frac{180}{2\pi} Z_0 \text{Re}(\sigma_{xy}), \tag{39}$$

where $t_\pm = E_\pm^t/|\boldsymbol{E}^i|$ are the transmission amplitudes in the circular polarization basis [83, 84] or chiral basis [85], $\boldsymbol{E}_\pm^t = E_x^t \pm \mathrm{i} E_y^t$. $\text{Re}(\sigma_{ij})$ means the real part of $\sigma_{ij}$ and $\arg(t_\pm)$ the complex argument. We have also provided the approximate expressions in the limit of weak absorption for isotropic materials. Note that, in this case, according to (39), the absorption peaks of $\text{Re}[\sigma_{xx}(\Omega)]$ shown in Figure 6, correspond to dips of the transmittance $\mathcal{T}$. Silicene and HgTe QWs have both longitudinal conductivities equal $\sigma_{xx} = \sigma_{yy}$, but this symmetry is broken for anisotropic materials like phosphorene [10, 85] (see later on Section 3.3). Therefore, in phosphorene, we cannot apply the approximation in eq.(39) and we have to use the strict equality.

In the circular polarization (right- and left-handed $\pm$) basis, the conductivity is given by $\sigma_\pm = \sigma_{xx} \pm \mathrm{i}\sigma_{xy}$, and the absorptive part is therefore $\text{Re}(\sigma_\pm) = \text{Re}(\sigma_{xx}) \mp \text{Im}(\sigma_{xy})$. In Appendix B we provide extra plots for the silicene conductivity under circular polarization which reproduce the results of [9].

## 3.1 Magneto-optical properties of graphene analogues

The current operator (35) for this case is $\boldsymbol{j} = (j_x, j_y) = ev(\xi\tau_x, \tau_y)$. The matrix elements

$$
\begin{aligned}
\langle \boldsymbol{m}|\tau_x|\boldsymbol{n}\rangle_{s\xi} &= A_m^{s\xi} B_n^{s\xi} \delta_{|m|-\xi,|n|} + A_n^{s\xi} B_m^{s\xi} \delta_{|m|+\xi,|n|} \, , \\
\langle \boldsymbol{m}|\tau_y|\boldsymbol{n}\rangle_{s\xi} &= -\mathrm{i} A_m^{s\xi} B_n^{s\xi} \delta_{|m|-\xi,|n|} + \mathrm{i} A_n^{s\xi} B_m^{s\xi} \delta_{|m|+\xi,|n|} \, ,
\end{aligned}
\tag{40}
$$

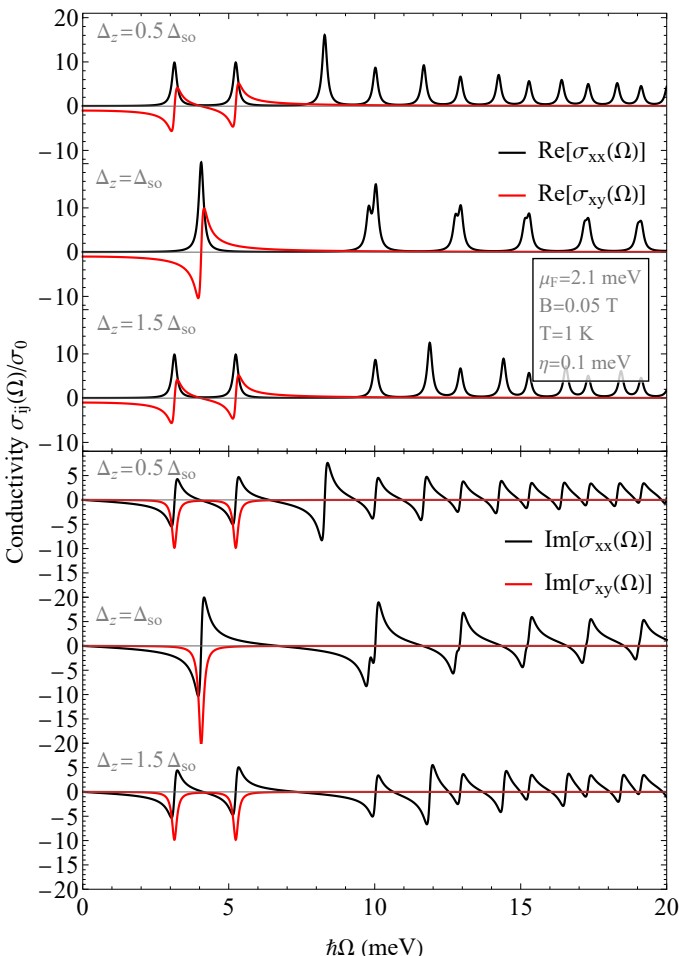

Figure 4: Real and imaginary parts of the longitudinal $\sigma_{xx}$ and transverse Hall $\sigma_{xy}$ magneto-optical conductivities in a silicene monolayer under three different electric potentials $\Delta_z = 0.5\Delta_{so}, \Delta_{so}, 1.5\Delta_{so}$, as a function of the polarized light frequency $\Omega$ and in $\sigma_0 = e^2/h$ units. We set the conductivity parameters as $\mu_F = 2.1$ meV, $B = 0.05$ T, $T = 1$ K and $\eta = 0.1$ meV.

provide the familiar selection rules $|n| = |m| \pm 1$ for LL transitions. Plugging (40) into the general expression (36) we obtain the magneto-optical conductivity for graphene analogues. In Figure 4 we plot the real and imaginary parts of the conductivity tensor components $\sigma_{ij}$ (in $\sigma_0 = e^2/h$ units) of silicene as a function of the polarized light frequency $\Omega$ at three different electric potentials $\Delta_z = 0.5\Delta_{so}, \Delta_{so}, 1.5\Delta_{so}$ around the critical point $\Delta_z^{(0)} = \Delta_{so}$, for a magnetic field $B = 0.05$ T and some representative values of the chemical potential $\mu_F = 2.1$ meV, temperature $T = 1$ K and scattering rate $\eta = 0.1$ meV. For $\hbar\Omega \in [0, 20]$ meV, we achieve convergence with 100 LLs, that is, restricting the sum in (36) as $\sum_{n=-\infty}^{\infty} \to \sum_{n=-100}^{100}$. More explicitly, for the parameters mentioned above,

$$\left| \sum_{n=-100}^{n=100} \sigma_{ij} - \sum_{n=-99}^{n=99} \sigma_{ij} \right| / \sigma_0 \leq \begin{cases} 10^{-5}, & \text{if } \sigma_{ij} = \text{Re}(\sigma_{xx}), \\ 10^{-15}, & \text{if } \sigma_{ij} = \text{Re}(\sigma_{xy}), \\ 10^{-3}, & \text{if } \sigma_{ij} = \text{Im}(\sigma_{xx}), \\ 10^{-14}, & \text{if } \sigma_{ij} = \text{Im}(\sigma_{xy}). \end{cases} \tag{41}$$

Each peak on the plot of the conductivity $\text{Re}(\sigma_{xx})$ against $\hbar\Omega$ represents an electron transi-

tion between two LLs $n, m$ connected by the selection rules $|n| = |m| \pm 1$ and generally arranged above and below the Fermi level $\mu_F$; this latter constrain comes from the Fermi functions factor $(f_m - f_n)$ of the Kubo formula (34), which becomes a step function at low temperatures. For more information, see the Supplemental Material [52] where we illustrate these electron transitions by arrows in the energy spectrum in an animated gif explained in Appendix D. The value of $\hbar\Omega$ where a peak of the conductivity occurs coincides with the energy difference $(E_n - E_m)$ of the LL transition $n \to m$. This is clear by looking at the denominator of the Kubo formula. For example, the two main peaks of $\mathrm{Re}(\sigma_{xx})$ at low frequencies $\hbar\Omega \in [2, 6]$ meV in Figure 4 correspond to the transitions $0 \to 1$ for spin and valley $s = \xi = 1$ and $s = \xi = -1$ (purple and green arrows in the animated gif of [52]). The other conductivity peaks located at higher frequencies correspond to electron transitions between higher LLs and different spin/valley combinations according to (40). When the external electric field $\Delta_z$ is such that the energy differences of the two main peaks are the same, that is, when $E_1^{++} - E_0^{++}$ is equal to $E_1^{--} - E_0^{--}$, both peaks merge into a bigger one. Using the silicene spectrum energy equation (7), we find that this condition is fulfilled at the critical point $\Delta_z = \Delta_{so}$ for any value of the magnetic field $B$. This result implies that we can extract information of the TPT occurring at $\Delta_z^{(0)} = \Delta_{so}$ by looking at the conductivity $\mathrm{Re}(\sigma_{xx})$ plot for different values of $\Delta_z$.

To be more specific, in Figure 5 we represent the behavior of the two observables given in (39), that is, the Faraday angle $\Theta_F$ and the transmittance $\mathcal{T}$, as a function of the polarized light frequency $\Omega$ around the critical point $\Delta_z^{(0)} = \Delta_{so} = 4.2$ meV. We focus on the frequency interval $\hbar\Omega \in [2, 6]$ meV where the main peaks (transition $0 \to 1$) in Figure 4 are located. We find an absolute minimum of the transmittance $\mathcal{T}_0 = 0.704$ at the critical point $\Delta_z^{(0)} = \Delta_{so}$ and $\hbar\Omega = 4.06$ meV. This "minimal" behavior does not depend on the particular values of magnetic field, chemical potential and temperature, which only change the actual value of $\mathcal{T}_0$ and $\hbar\Omega$ of the peak. Actually, the minimum peaks in the transmittance plot are related to the maximum peaks of the absortance $\mathrm{Re}(\sigma_{xx})$, according to equation (39). The Faraday angle at the critical point (black curve in Figure 5) changes sign at the minimum transmittance point $\hbar\Omega = 4.06$ meV, a behavior that can also be extrapolated to other 2D materials (se later for HgTe QWs and phosphorene). In fact, each peak of the transmittance in Figure 5 coincides in frequency with an inflection point of the Faraday angle, where it attains a value of 0 degrees.

Changing the chemical potential $\mu_F$ locks/unlocks other electronic transitions, so we would see different peaks in the conductivity and transmittance plots (see e.g., [10]). Increasing the scattering rate $\eta$ smoothes the peaks in the transmittance, so it would be more difficult to distinguish when they overlap. We have choosen values of $\eta$ approximately an order of magnitude below the frequency of the conductivity peaks, for which the resolution is fine.

For completeness, in Appendix E we show several contour plots of the Faraday angle using different cross sections in the $\{\hbar\Omega, \Delta_z, B, T, \mu_F\}$ parameter space.

## 3.2 Magneto-optical properties of zincblende heterostructures

From the Hamiltonian (10), the current operator (35) for zincblende heterostructures is

$$
\begin{aligned}
j_x^s &= \frac{e}{\hbar} \left( s\alpha\tau_x - 2k_x(\beta\tau_z + \delta\tau_0) \right), \\
j_y^s &= \frac{e}{\hbar} \left( \alpha\tau_y - 2k_y(\beta\tau_z + \delta\tau_0) \right),
\end{aligned}
\tag{42}
$$

which, after minimal coupling according to the general prescription (3), results in

$$
\begin{aligned}
j_x^s &= \frac{e}{\hbar} \left( s\alpha\tau_x - \sqrt{2}\frac{a^\dagger + a}{\ell_B}(\beta\tau_z + \delta\tau_0) \right), \\
j_y^s &= \frac{e}{\hbar} \left( \alpha\tau_y + i\sqrt{2}\frac{a^\dagger - a}{\ell_B}(\beta\tau_z + \delta\tau_0) \right).
\end{aligned}
\tag{43}
$$

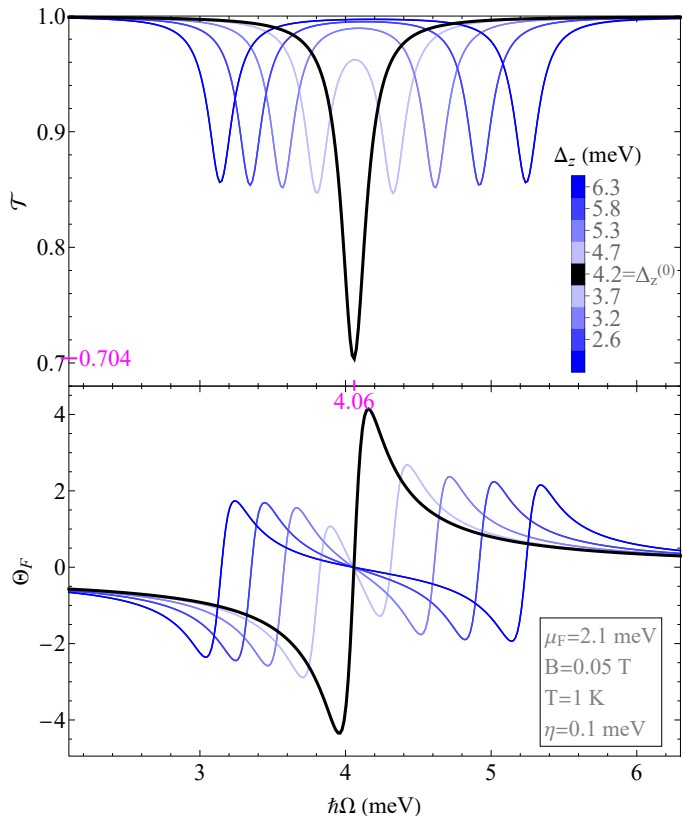

Figure 5: Transmittance $\mathcal{T}$ and Faraday angle $\Theta_{\mathrm{F}}$ (in degrees) in a silicene monolayer as a function of the incident polarized light frequency $\Omega$, and for different electric fields below and above the critical (black line) electric field $\Delta_z^{(0)} = \Delta_{\mathrm{so}} = 4.2\,\mathrm{meV}$. $\mathcal{T}$ and $\Theta_{\mathrm{F}}$ are symmetric about $\Delta_z^{(0)}$. We set the conductivity parameters as $\mu_{\mathrm{F}} = 2.1\,\mathrm{meV}$, $B = 0.05\,\mathrm{T}$, $T = 1\,\mathrm{K}$ and $\eta = 0.1\,\mathrm{meV}$.

Note that, in fact, $j_y^s$ does not depend on $s$. The current matrix elements for this case are

$$
\begin{aligned}
\langle \boldsymbol{m} | j_x^s | \boldsymbol{n} \rangle_s &= \frac{es\alpha}{\hbar} \Xi_{m,n}^{s,+} - \frac{\sqrt{2}e}{\hbar\ell_B} \Phi_{m,n}^{s,+}, \\
\langle \boldsymbol{m} | j_y^s | \boldsymbol{n} \rangle_s &= -\mathrm{i}\frac{e\alpha}{\hbar} \Xi_{m,n}^{s,-} + \mathrm{i}\frac{\sqrt{2}e}{\hbar\ell_B} \Phi_{m,n}^{s,-},
\end{aligned}
\tag{44}
$$

where

$$
\begin{aligned}
\Xi_{m,n}^{s,\pm} &= \left( A_m^s B_n^s \delta_{|m|-s,|n|} \pm A_n^s B_m^s \delta_{|m|+s,|n|} \right), \\
\Phi_{m,n}^{s,\pm} &= \left( (\delta+\beta) A_m^s A_n^s + (\delta-\beta) B_m^s B_n^s \right) \left( \sqrt{|n|+1+\tfrac{s-1}{2}}\, \delta_{|m|-1,|n|} \pm \sqrt{|n|-\tfrac{s+1}{2}}\, \delta_{|m|+1,|n|} \right).
\end{aligned}
\tag{45}
$$

Despite the more involved structure of the current than for silicene, the corresponding matrix elements maintain the same familiar selection rules $|n| = |m| \pm 1$ for LL transitions.

Inserting the matrix elements (40) into the general expression (36) we obtain the magneto-optical conductivity for general zincblende heterostructures. In Figure 6 we plot the real and imaginary parts of the conductivity tensor components $\sigma_{ij}$ (in $\sigma_0 = e^2/h$ units) of a HgTe QW as a function of the polarized light frequency $\Omega$ at three different HgTe layer thicknesses $\lambda = 5.50\,\mathrm{nm} < \lambda_c$, $\lambda = 6.17\,\mathrm{nm} = \lambda_c$, and $\lambda = 7.00\,\mathrm{nm} > \lambda_c$, a magnetic field $B = 0.5\,\mathrm{T}$

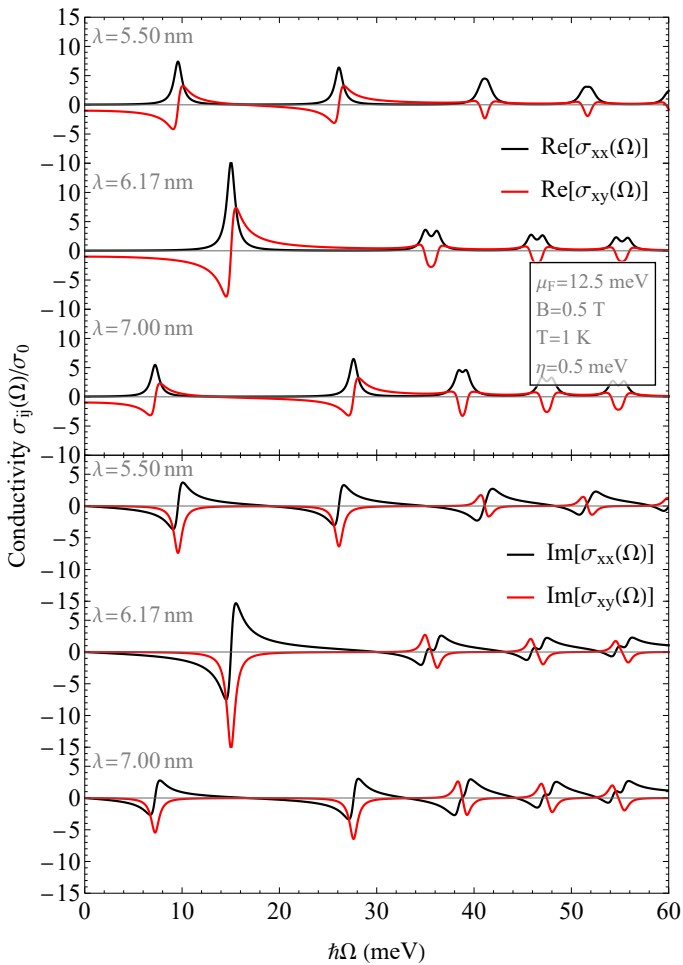

Figure 6: Real and imaginary parts of the longitudinal $\sigma_{xx}$ and transverse Hall $\sigma_{xy}$ magneto-optical conductivities in a bulk HgTe QW of thickness $\lambda = 5.50, 6.17, 7.00$ nm, as a function of the polarized light frequency $\Omega$ and in $\sigma_0 = e^2/h$ units. We set the conductivity parameters as $\mu_F = 12.5$ meV, $B = 0.5$ T, $T = 1$ K and $\eta = 0.5$ meV.

and some representative values of the chemical potential $\mu_F = 12.5$ meV, temperature $T = 1$ K and scattering rate $\eta = 0.5$ meV. For $\hbar\Omega \in [0, 60]$ meV, we achieve convergence with 100 LLs, that is, restricting the sum in (36) as $\sum_{n=-\infty}^{\infty} \rightarrow \sum_{n=-100}^{100}$. More explicitly, for the parameters mentioned above,

$$\left| \sum_{n=-100}^{n=100} \sigma_{ij} - \sum_{n=-99}^{n=99} \sigma_{ij} \right| / \sigma_0 \leq \begin{cases} 10^{-5}, & \text{if } \sigma_{ij} = \text{Re}(\sigma_{xx}), \\ 10^{-4}, & \text{if } \sigma_{ij} = \text{Re}(\sigma_{xy}), \\ 10^{-3}, & \text{if } \sigma_{ij} = \text{Im}(\sigma_{xx}), \\ 10^{-7}, & \text{if } \sigma_{ij} = \text{Im}(\sigma_{xy}). \end{cases} \tag{46}$$

Similar to silicene, we can see in Figure 6 that there are multiple peaks in the absorptive components $\text{Re}(\sigma_{xx})$ and $\text{Im}(\sigma_{xy})$, corresponding to transitions between occupied and unoccupied LLs obeying the selection rules $|n| = |m| \pm 1$. At lower frequencies $\hbar\Omega \in [0, 30]$ meV, inside each curve of Figure 6, we find the main peaks corresponding to the transitions $0 \rightarrow 1$ for spin $s = 1$ and $s = -1$. Both peaks merge approximately at $\lambda \simeq \lambda_c = 6.17$ nm. This is because the energy differences $E_1^+ - E_0^+$ and $E_1^- - E_0^-$ are similar when $\lambda \simeq \lambda_c$ for low magnetic

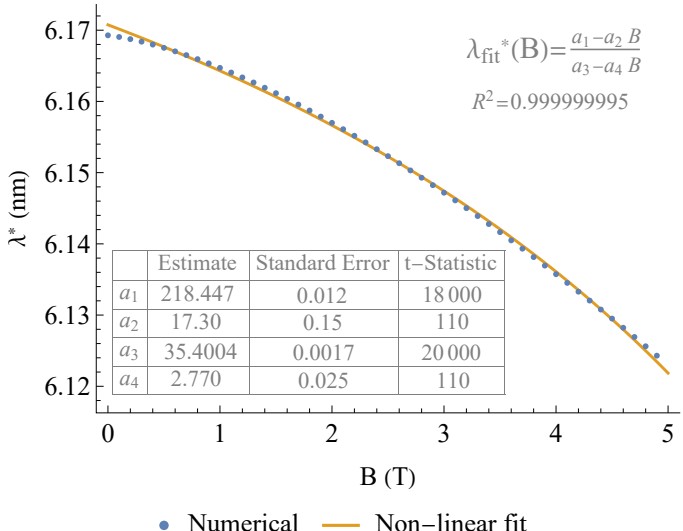

Figure 7: Numerical solutions $\lambda^*$ (in nm, blue dots) of the equation $E_1^+ - E_0^+ = E_1^- - E_0^-$ (energies (16,17) of HgTe QW) for 50 different values of the external magnetic field $B$. In orange, non-linear fit (47) of the numerical values.

fields $B \ll 1$ T, according to equations (16,17). In order to extend this result to higher values of the magnetic field, we insert the parameter fits (22) into the equation $E_1^+ - E_0^+ = E_1^- - E_0^-$, and solve it numerically for $\lambda^* = \lambda^*(B)$, obtaining the values represented by blue dots in Figure 7. These values fit the equation

$$\lambda_{\text{fit}}^*(B) = \frac{218.4 - 17.3B}{35.4 - 2.8B}, \tag{47}$$

which is represented as an orange curve in Figure 7. Consequently, only for small magnetic fields, we can infer the critical thickness $\lambda_c$ where the TPT in HgTe QW occurs from the conductivity $\text{Re}(\sigma_{xx})$ plot, that is, $\lambda^* \simeq \lambda_c = 6.17$ nm for $B \ll 1$ T.

The behavior of the Faraday angle and the transmittance as a function of the polarized light frequency $\Omega$ around the critical HgTe layer thickness $\lambda_c = 6.17$ nm (at which the material parameter $\mu$ changes sign/Chern number) is shown in Figure 8. As for silicene, we focus on the lower frequencies $\hbar\Omega \in [0, 30]$ meV where the main peaks are located, and find again a minimum of the transmittance, this time $\mathcal{T}_0 = 0.78$, at the critical point $\lambda_c$ and $\hbar\Omega = 15.0$ meV. For this material, the "minimal" behavior does depend on the particular values of magnetic field, as we saw in equation (47). However, for small magnetic fields like $B = 0.5$ T in Figure 8, the minimum of the transmittance still takes place at $\lambda^* \simeq \lambda_c = 6.17$ nm. The Faraday angle at the critical point (black curve in Figure 8) changes sign at the minimum transmittance frequency $\hbar\Omega = 15.0$ meV, a behavior shared with silicene.

For completeness, in Appendix E we show several contour plots of the Faraday angle using different cross sections in the $\{\hbar\Omega, \lambda, B, T, \mu_F\}$ parameter space.

## 3.3 Magneto-optical properties of phosphorene and effect of anisotropies

From the phosphorene Hamiltonian (25), the current operator (35) is

$$j_x^s = \frac{e}{\hbar}\left(\gamma\tau_x + k_x(\tau_0(\alpha_x - \beta_x) + \tau_z(\alpha_x + \beta_x))\right),$$
$$j_y^s = \frac{e}{\hbar}k_y\left(\tau_0(\alpha_y - \beta_y) + \tau_z(\alpha_y + \beta_y)\right), \tag{48}$$

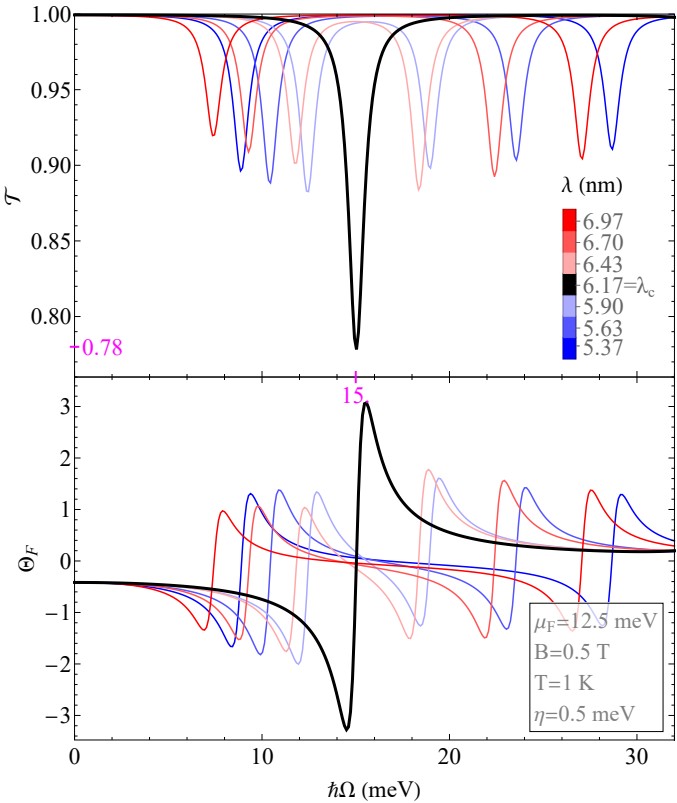

Figure 8: Transmittance $\mathcal{T}$ and Faraday angle $\Theta_{\mathrm{F}}$ (in degrees) in a bulk HgTe QW as a function of the polarized light frequency $\Omega$, and for thickness $\lambda < \lambda_c, \lambda = \lambda_c$ and $\lambda > \lambda_c$, with $\lambda_c = 6.17$ nm (black line). We set the conductivity parameters as $\mu_{\mathrm{F}} = 12.5$ meV, $B = 0.5$ T, $T = 1$ K and $\eta = 0.5$ meV.

which, after minimal coupling, according to prescription (29), results in

$$
\begin{aligned}
j_x^s &= \frac{e}{\hbar}\left(\gamma\tau_x + \frac{a^\dagger + a}{\sqrt{2}\alpha_{yx}\ell_B}(\tau_0(\alpha_x - \beta_x) + \tau_z(\alpha_x + \beta_x))\right), \\
j_y^s &= \frac{e}{\hbar}\frac{\alpha_{yx}(a^\dagger - a)}{\mathrm{i}\sqrt{2}\ell_B}\left(\tau_0(\alpha_y - \beta_y) + \tau_z(\alpha_y + \beta_y)\right).
\end{aligned}
\tag{49}
$$

Plugging these matrix elements into the general expression (36) we obtain the magneto-optical conductivity for phosphorene. Note that, unlike silicene and HgTe QW, there is now a large asymmetry between $\sigma_{xx}$ and $\sigma_{yy}$ (about one order of magnitude difference), as evidenced by Figure 9. This asymmetry was already highlighted by [71], where tunable optical properties of multilayer black phosphorus thin films were studied for $B = 0$. In Figure 9 we plot the real and imaginary parts of the conductivity tensor components $\sigma_{ij}$ (in $\sigma_0 = e^2/h$ units) of phosphorene as a function of the polarized light frequency $\Omega$, for some values of the electric potential around $\Delta_z^{(0)} = -E_g = -1.52$ eV (closing the energy gap), a magnetic field of $B = 0.5$ T, like in Figure 3, and some representative values of the chemical potential $\mu_{\mathrm{F}} = -0.417$ eV, temperature $T = 1$ K and scattering rate $\eta = 0.2$ meV. We are using the same threshold of $N = 1000$ Fock states that we used to find convergence in the first 6 Hamiltonian eigenstates of the numerical diagonalization in Figure 3. This convergence is ensured for $\hbar\Omega \in [0, 20]$ meV. The anisotropic character of phosphorene also implies that the current

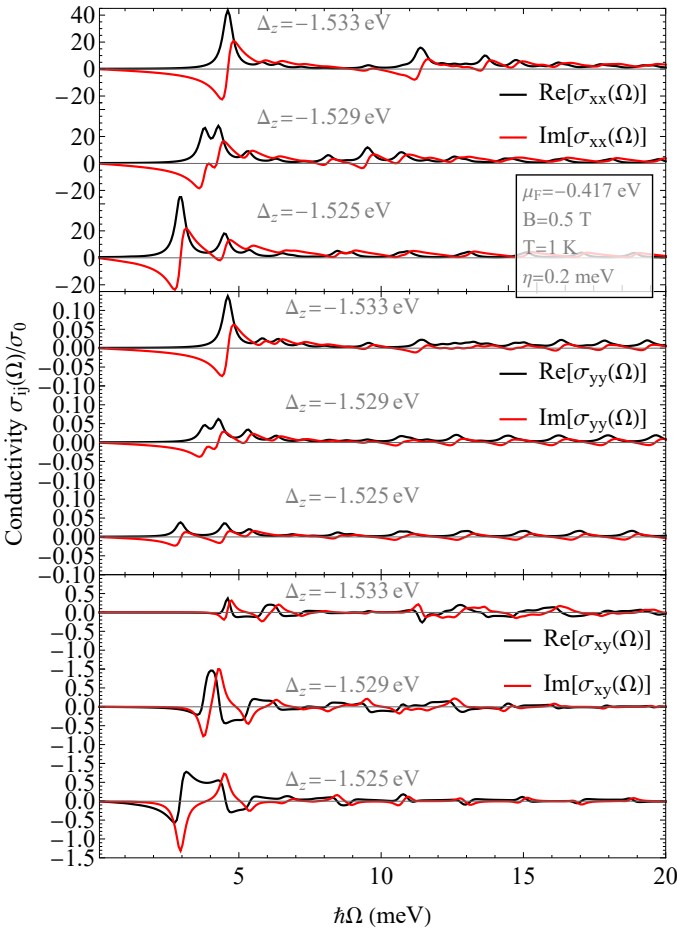

Figure 9: Real and imaginary parts of the longitudinal $\sigma_{xx}, \sigma_{yy}$ and transverse Hall $\sigma_{xy}$ magneto-optical conductivities in a phosphorene monolayer, as a function of the polarized light frequency $\Omega$ and in $\sigma_0 = e^2/h$ units. Phosphorene is under a perpendicular electric field potential $\Delta_z^{(0)} = -E_g = -1.52$ eV closing the energy gap in Figure 3. The $y$-axis ticks have different values in each subplot as the conductivities $\sigma_{xy}$ and $\sigma_{yy}$ attain smaller values than $\sigma_{xx}$ (phosphorene anisotropy). We set the conductivity parameters as $\mu_F = -0.417$ meV, $B = 0.5$ T, $T = 1$ K and $\eta = 0.2$ meV.

$j_y^s$ is significantly lower than $j_x^s$ [the Hamiltonian (25) is of second order in $k_y$]. This makes transversal components of the conductivity significantly lower than longitudinal components. This is why we have disposed Figure 9 in a slightly different manner from Figures 4 for silicene and 6 for HgTe QW, which display a more isotropic structure.

Due to the parity symmetry of the Hamiltonian (30), only the electronic transitions between LLs of different parities are allowed [30]. The main peak (smaller frequency) of the conductivity $\text{Re}(\sigma_{xx})$ in Figure 9 corresponds to the electronic transitions $E_0^{\text{even}} \to E_3^{\text{odd}}$ and $E_1^{\text{odd}} \to E_2^{\text{even}}$, which have approximately the same energy difference for all $\Delta_z < -1.53$ eV with a tolerance $\leq 10^{-14}$ eV. That is, $E_0^{\text{even}}$ and $E_1^{\text{odd}}$, and $E_2^{\text{even}}$ and $E_3^{\text{odd}}$, are degenerate for all $\Delta_z < -1.53$ eV as the spectrum in Figure 3 shows. When the degeneration is broken around the electric potential $\Delta_z \simeq -1.53$ eV, the main conductivity $\text{Re}(\sigma_{xx})$ peak splits into two as we can see in Figure 9.

The anisotropic character of phosphorene also affects the Faraday angle, which attains much lower values (in absolute value) than for silicene or HgTe QWs. Indeed, in Figure 10 we plot Faraday angle and transmittance as a function of the polarized light frequency $\Omega$ for

different electric field potentials $-1.535 \leq \Delta_z \leq -1.519$ eV. Like for silicene and HgTe QWs, we find a minimal behavior in the transmittance of phosphorene $\mathcal{T}_0 = 0.50$ for a polarized light frequency $\hbar\Omega = 2.6$ meV at electric field potential $\Delta_z^{(0)} = -1.523$ eV, which is close to minus the energy gap $-E_g = -1.52$ eV. Note that this value of the minimal transmittance of phosphorene is much smaller than for silicene and HgTe QWs; actually, the assumption of low absortance in formula (39) is no longer valid here and we have used the exact expressions for $\mathcal{T}$ and $\Theta_F$ in (39). Moreover, unlike for graphene analogues and HgTe QWs, this minimum of the transmittance does not seem to be related to the union of two conductivity peaks into a bigger one; rather, it is simply related to the energy gap closure. Actually, the critical electric potential $\Delta_z^{(0)}$ where the transmittance of phosphorene reaches a minimum depends on the magnetic field $B$ chosen, as Figure 11 shows. We perform a non-linear fit of the numerical values of $\Delta_z^{(0)}(B)$ and obtain the equation ($B$ in dimensionless units)

$$\left(\Delta_z^{(0)}\right)_{\text{fit}}(B) = \frac{-77.4 - 3.5B}{50.9 + 2.2B} \text{ eV}, \tag{50}$$

which is represented as a orange curve in Figure 11. For small magnetic fields, we can deduce that the critical electric field potential is similar to minus the energy gap $-E_g$ of phosphorene, that is $\Delta_z^{(0)}(B) \simeq -E_g = -1.52$ eV for $B \ll 1$ T. We have also checked numerically that the critical electric potentials $\Delta_z^{(0)}(B)$ are independent of the parameters $\mu_F$ and $\eta$ for a fixed magnetic field $B$. However, we set different values of $\mu_F$ for small fields $B \leq 2$ T (see caption of Figure 11), in order to avoid blocking the electric transition $E_1^{\text{odd}} \to E_2^{\text{even}}$ of the main peak of the transmittance. We also increment $N$ as $B$ decreases in order to achieve convergence in the diagonalization.

Additionally, Figure 10 shows how one peak of the transmittance splits into two around $\Delta_z \simeq -1.53$ eV (blue lines), since the LL $E_0^{\text{even}}$ breaks its degeneration approximately for $\Delta_z > -1.53$ eV (see Figure 3). For $\Delta_z = \Delta_z^{(0)} = -1.523$ eV (thick black line), the big peak on the left in Figure 10 corresponds to the electronic transition $E_1^{\text{odd}} \to E_2^{\text{even}}$, and moves toward smaller values of $\hbar\Omega$ when increasing $\Delta_z$. The other small peak in the black line corresponds to the electronic transition $E_0^{\text{even}} \to E_3^{\text{odd}}$, which moves toward bigger values of $\hbar\Omega$ when increasing $\Delta_z$. The Faraday angle also presents inflection points at the frequencies where the peaks of the transmittance are located.

Therefore, we see that anisotropies affect the values of the Faraday angle and transmittance. There are mechanical ways of introducing anisotropies in 2D materials by subjecting them to strain (like for strained [86] or rippled [87] graphene). This kind of anisotropies can be treated by replacing the scalar Fermi velocity $v$ by a $2\times2$ symmetric tensor $\boldsymbol{v}$ (see e.g. [82]). Namely, for graphene, the Hamiltonian (1) vector $\boldsymbol{d}$ components $d_j = \hbar v k_j$ are replaced by $d_j = \hbar k_i v_{ij}, i = 1, 2, d_3 = 0$. Actually, for uniformly strained graphene with strain tensor $\boldsymbol{\varepsilon}$, the Fermi velocity tensor is (up to first order) $\boldsymbol{v} = v(\tau_0 - \beta\boldsymbol{\varepsilon})$ (see e.g. [82,88]), where $\beta \sim 2$. The relation between the isotropic $\boldsymbol{\sigma}^0$ and the anisotropic $\boldsymbol{\sigma}$ magneto-optical conductivity tensors is simply $\boldsymbol{\sigma}(\Omega, B) = \boldsymbol{v}\boldsymbol{\sigma}^0(\Omega, \mathcal{B})\boldsymbol{v}/\det(\boldsymbol{v})$, with $\mathcal{B} = B\det(\boldsymbol{v})/v^2$ an effective magnetic field. Interesting discussions on how measurements of dichroism and transparency for two different light polarization directions can be used to determine the magnitude and direction of strain can be found in [81]. Also, photoelastic effects in graphene [86], strain-modulated anisotropies in silicene [89,90], etc. The band gap $E_g = E_v - E_c$ of phosphorene can be furthermore modulated by strain and by the number of layers in a stack [60,70].

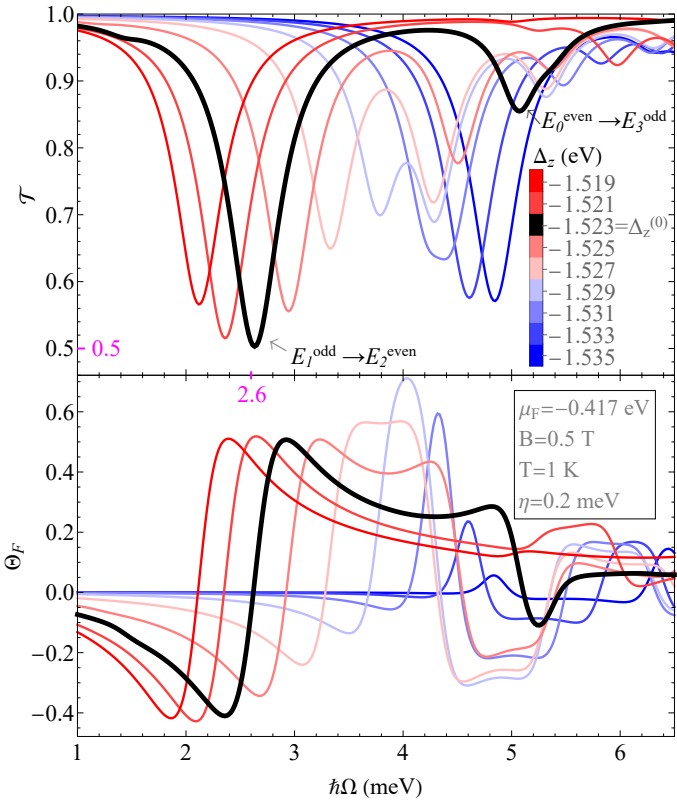

Figure 10: Transmittance $\mathcal{T}$ and Faraday angle $\Theta_F$ (in degrees) in a phosphorene monolayer as a function of the polarized light frequency $\Omega$, and for electric fields $-1.535 < \Delta_z < -1.519$ eV around the minus energy gap $-E_g = -1.52$ eV. The black line corresponds to the electric potential $\Delta_z^{(0)} = -1.523$ eV $\simeq -E_g$ where the transmittance attains a minimum of $\mathcal{T}_0 = 0.5$ at $\hbar\Omega = 2.5$ meV. We set the conductivity parameters as $\mu_F = -0.417$ meV, $B = 0.5$ T, $T = 1$ K and $\eta = 0.2$ meV.

## 4  Conclusions

We have studied magneto-optical properties of different 2D materials, focusing on transmittance and Faraday rotation near the critical point of the topological phase transition for topological insulators like silicene and HgTe quantum wells. We have seen that, in all topological 2D materials analyzed, transmittance attains an absolute minimum $\mathcal{T}_0$ at the critical TPT point for a certain value $\Omega_0$ of the normal incident polarized light frequency. This is a universal behavior for graphene analogues, that is, the minimal behavior of the transmittance does not depend on the chosen values of magnetic field, chemical potential and temperature, although the location of $\Omega_0$ varies with them. In addition, we have found that each peak of the transmittance coincides in frequency with an inflection point of the Faraday angle, for a fixed selection of the electric field, magnetic field, chemical potential and temperature parameters.

This extremal universal behavior is shared with other topological 2D materials like HgTe quantum wells as long as the applied magnetic field remains small enough $B \ll 1$ T. In HgTe quantum wells we have verified that there is a minimum of the transmittance $\mathcal{T}_0$ at the critical HgTe layer thickness at a given frequency $\Omega_0'$ (for this material this minimal behavior depends on the magnetic field) and the Faraday angle at the critical point changes sign at the minimum transmittance frequency $\Omega_0'$.

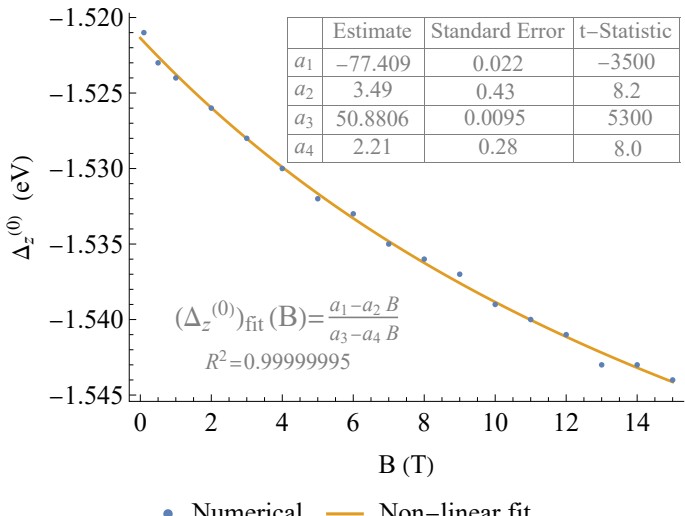

| | Estimate | Standard Error | t−Statistic |
|---|---|---|---|
| $a_1$ | −77.409 | 0.022 | −3500 |
| $a_2$ | 3.49 | 0.43 | 8.2 |
| $a_3$ | 50.8806 | 0.0095 | 5300 |
| $a_4$ | 2.21 | 0.28 | 8.0 |

$$(\Delta_z^{(0)})_{\text{fit}}(B) = \frac{a_1 - a_2 B}{a_3 - a_4 B}$$

$$R^2 = 0.99999995$$

• Numerical — Non−linear fit

Figure 11: Electric field potential at which phosphorene transmittance reaches a minimum, as a function of different magnetic fields. In orange, non-linear fit (50) of the numerical values. In general, we set the conductivity parameters $\mu_F = -0.41$ eV, $T = 1$ K, $\eta = 1$ meV, and use $N = 300$ Fock state in the numerical diagonalization, for all $B \geq 3$ T. For smaller magnetic fields $B = 0.1, 0.5, 1, 2$ T, we set $\mu_F = -0.419, -0.418, -0.416, -0.416$ eV respectively. In the case of $B = 0.1$ T we also set $\eta = 1$ meV and $N = 500$ Fock states to achieve energy diagonalization convergence.

For other non-topological anisotropic materials like phosphorene, this minimal behavior of the transmittance still remains when the energy gap is closed, the Faraday angle being much smaller (in absolute value) than in silicene and HgTe QWs. In this case the critical electric potential where the transmittance reaches a minimum depends on the magnetic field.

Therefore, these extremal properties of transmittance/absortance and chirality change of Faraday angle at the critical point turn out to provide sharp markers of either the topological phase transition or the energy gap closure.

## Acknowledgments

**Funding information** We thank the support of Grant PID2022-138144NB-I00 funded by Spanish MICIU and Junta de Andalucía through the projects FEDER/UJA-1381026 and FQM-381. AM thanks the Spanish MIU for the FPU19/06376 predoctoral fellowship. OC is on sabbatical leave at Granada University, Spain, since the 1st of September 2023. OC thanks support from the program PASPA from DGAPA-UNAM.

## A  Landau levels plot versus external magnetic field

We provide an additional plot of the Landau levels of the three different materials as a function of the external magnetic field $B$. Critical values of the electric field and layer thickness are selected, that is, in the case of the silicene $\Delta_z = \Delta_{so} = 4.2$ meV, for the HgTe QW $\lambda = \lambda_c = 6.17$ nm, and for the phosphorene $\Delta_z = -E_g = -1.52$ eV.

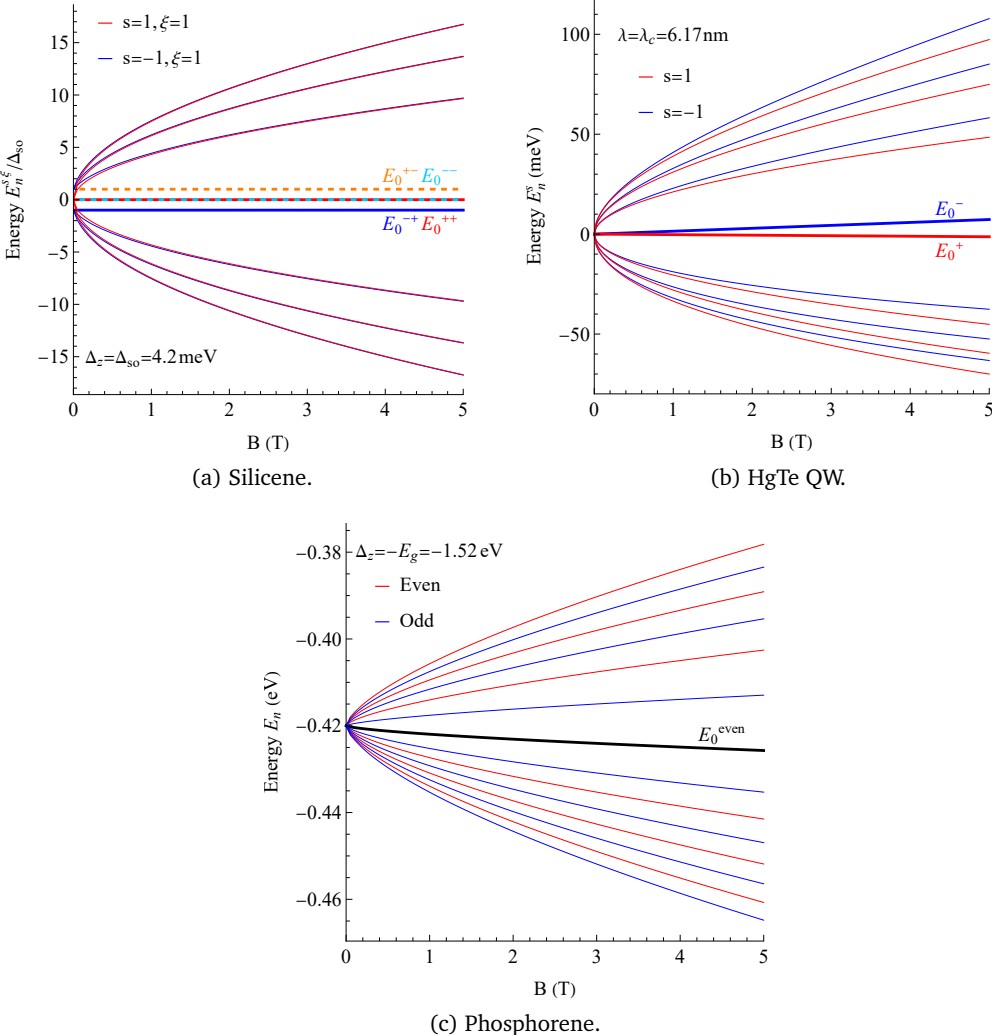

Figure 12: Energies (Landau levels) of (a) Silicene, (b) HgTe QW, and (c) Phosphorene as a function of the external magnetic field $B$. Critical values of the electric field and layer thickness are selected, that is, in the case of the silicene $\Delta_z = \Delta_{so} = 4.2$ meV, for the HgTe QW $\lambda = \lambda_c = 6.17$ nm, and for the phosphorene $\Delta_z = -E_g = -1.52$ eV.

# B  Silicene conductivity in the circularly polarization basis

We complete the analysis of magneto-optical properties of graphene analogues by discussing the case of circularly polarized light. In this case, the conductivity is $\sigma_{\pm}(\Omega) = \sigma_{xx}(\Omega) \pm i\sigma_{xy}(\Omega)$ for right-handed (+) and left-handed (-) polarization [91]. Therefore, the absorptive part is $\mathrm{Re}(\sigma_{\pm}) = \mathrm{Re}(\sigma_{xx}) \mp \mathrm{Im}(\sigma_{xy})$. In Figure 13, we present both absorptive parts $\mathrm{Re}(\sigma_{\pm})$ for a silicene monolayer under an electric potential $\Delta_z = 0.5\Delta_{so}$ as a function of the frequency of the incident light $\Omega$. The conductivity parameters are specifically chosen to reproduce the results in [9], that is, $\mu_F = 3.0\Delta_{so}$, $B/\Delta_{so}^2 = 657$ G/meV$^2$, $T = 0$ K and $\eta = 0.05\Delta_{so}$. Note that we have defined the conductance quantum as $\sigma_0 = e^2/h = 38.8\,\mu$S, whereas the authors in reference [9] take $\sigma_0 = e^2/(4\hbar)$.

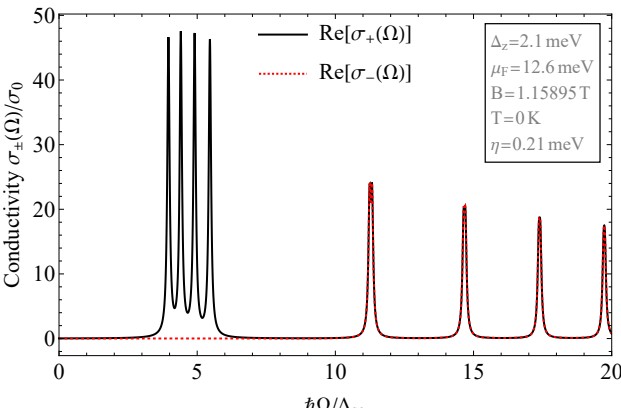

Figure 13: Conductivity absorptive parts $\mathrm{Re}(\sigma_{\pm}) = \mathrm{Re}(\sigma_{xx}) \mp \mathrm{Im}(\sigma_{xy})$ for right-handed (+) and left-handed (-) polarization in a silicene monolayer under an electric potential $\Delta_z = 0.5\Delta_{\mathrm{so}}$, as a function of the polarized light frequency $\Omega$ (in $\sigma_0 = e^2/h$ units). We set the conductivity parameters $\mu_F = 3.0\Delta_{\mathrm{so}}$, $B/\Delta_{\mathrm{so}}^2 = 657$ G/meV$^2$, $T = 0$ K and $\eta = 0.05\Delta_{\mathrm{so}}$ as in Ref. [9].

## C  HgTe quantum well conductivity with Zeeman effect

We recalculate the conductivity of the HgTe quantum well with and without Zeeman coupling to support the argument that the results are qualitatively equivalent, the quantitative differences being small. A layer thickness of $\lambda = 7.0$ nm is selected, so the material parameters are $\alpha = 365$ meV·nm, $\beta = -686$ meV·nm$^2$, $\delta = -512$ meV·nm$^2$, and $\mu = -10$ meV, as taken from Ref. [49]. In Figure 14, we plot the real and imaginary parts of the longitudinal $\sigma_{xx}$ and transverse $\sigma_{xy}$ conductivities as a function of the polarized light frequency $\Omega$. The conductivity parameters are chosen to reproduce the results in [17] with Zeeman coupling, that is, $\mu_F = 8$ meV, $B = 5$ T, $T = 1$ K and $\eta = 1$ meV. The conductance quantum used here is again $\sigma_0 = e^2/h = 38.8\,\mu$S, whereas the authors in reference [17] take $\sigma_0 = e^2/\hbar$.

## D  Animations of the energy spectrum and conductivities

Attached in the Supplemental Material [52] is a series of animations called:

```
-Silicene_Conductivity_and_Energy_VS_Omega.gif,
-HgTe_Conductivity_and_Energy_VS_Omega.gif,
-Phosphorene_Conductivity_and_Energy_VS_Omega.gif,
```

where we plot the energy spectrum at right, and the real part $\mathrm{Re}[\sigma_{xx}(\Omega)]$ and $\mathrm{Re}[\sigma_{xy}(\Omega)]$ of the conductivity components at left, for three different materials studied in the main text: silicene, HgTe QW, and phosphorene. The external electric field $\Delta_z$ in the case of the silicene and phosphorene, and the layer thickness $\lambda$ of the HgTe QW, are used as "time coordinate" on the animations, so each frame corresponds to one value of these control parameters.

The conductivities are plotted as a function of the polarized light frequency $\Omega$, and they change in each frame according to the values of $\Delta_z$ or $\lambda$. Therefore, we can observe how the main peaks of the longitudinal conductivity $\mathrm{Re}(\sigma_{xx})$ merge for the critical values $\Delta_z^{(0)} = \Delta_{\mathrm{so}} = 4.12$ meV (silicene) or $\lambda = \lambda_c = 6.17$ nm (HgTe QW), where the topological phase transition occurs in these 2D materials.

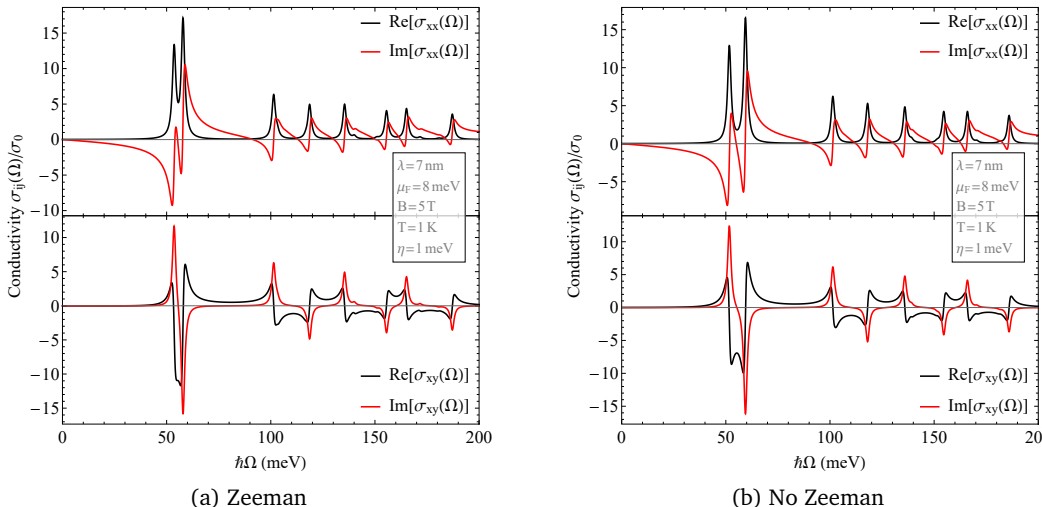

(a) Zeeman                                    (b) No Zeeman

Figure 14: Real and imaginary parts of the longitudinal $\sigma_{xx}$ and transverse Hall $\sigma_{xy}$ (magneto-)optical conductivities in a bulk HgTe QW of a thicknesses $\lambda = 7.0$ nm, as a function of the polarized light frequency $\Omega$ (in $\sigma_0 = e^2/h$ units) with and without Zeeman coupling. We set the conductivity parameters $\mu_F = 8$ meV, $B = 5$ T, $T = 1$ K and $\eta = 1$ meV, as in Ref. [17].

In the case of the phosphorene, we only observe the degeneration of Landau levels $n = 0$ and $n = 1$ in the conductivity around the electric potential $\Delta_z \simeq -1.53$ eV. That is, the electronic transitions $E_1^{\text{odd}} \to E_2^{\text{even}}$ and $E_0^{\text{even}} \to E_3^{\text{odd}}$ have a similar energy and share a longitudinal conductivity peak (main peak at left in the gif), until the degeneration breaks for electric fields approximately higher than $-1.53$ eV, when both electronic transitions will have different energies so the main peak will split into two.

On the other hand, the energy spectrum is static on the animation, as it is plotted as a function of all the values that $\Delta_z$ or $\lambda$ take. However, we plot a moving vertical dashed line on it, representing the value of $\Delta_z$ or $\lambda$ in the conductivity frame. On top of this vertical line, we also draw arrows representing the electronic transitions allowed between Landau levels (LLs) for the specific value $\Delta_z$ or $\lambda$, where the Fermi energy $\mu_F$ is represented by an horizontal dashed line. The color of the arrows is the same as the color of the points plotted on the top of the longitudinal conductivity main peaks. The length of the arrows represents the energy difference $|E_n - E_m|$ between the corresponding Landau levels in this particular electronic transition $n \leftrightarrow m$, which also coincides with the frequency $\hbar\Omega$ of the longitudinal conductivity peak associated with this transition. Therefore, when two arrows have the same length, we can observe two longitudinal conductivity peaks merging at the critical point. We have only drawn the arrows of the main peaks or lower Landau level electronic transitions for the sake of simplicity.

# E   Faraday angle contour plots

For completeness, in Figure 15 we show the variability of the Faraday angle for silicene across the parameter space: polarized light frequency $\hbar\Omega$, electric field potential $\Delta_z$, magnetic field $B$, temperature $T$ and chemical potential $\mu_F$}, using several contour plots corresponding to different cross sections. Also, in Figure 16 we do the same for the Faraday angle in HgTE quantum wells using different cross sections in the $\{\hbar\Omega, \lambda, B, T, \mu_F\}$ parameter space, where

the critical thickness $\lambda_c \simeq 6.17$ nm is marked with a vertical magenta grid line. The variability of the Faraday angle with those parameters is shown with a color code (in degrees), going from the most negative value (blue) to the most positive (red).

In the case of the silicene, we have also repeated the contour plot of the parameters $\{\hbar\Omega, \Delta_z\}$ for different values of the temperature $T = 1, 10, 100, 200$ K in Figure 17. The shape of the contour lines is almost the same when varying the temperature, but oscillation amplitude in the Faraday angle diminish when increasing $T$, as the colors of the plots tend to be more flat and yellow ($\Theta_F \simeq 0$ according to the legend).

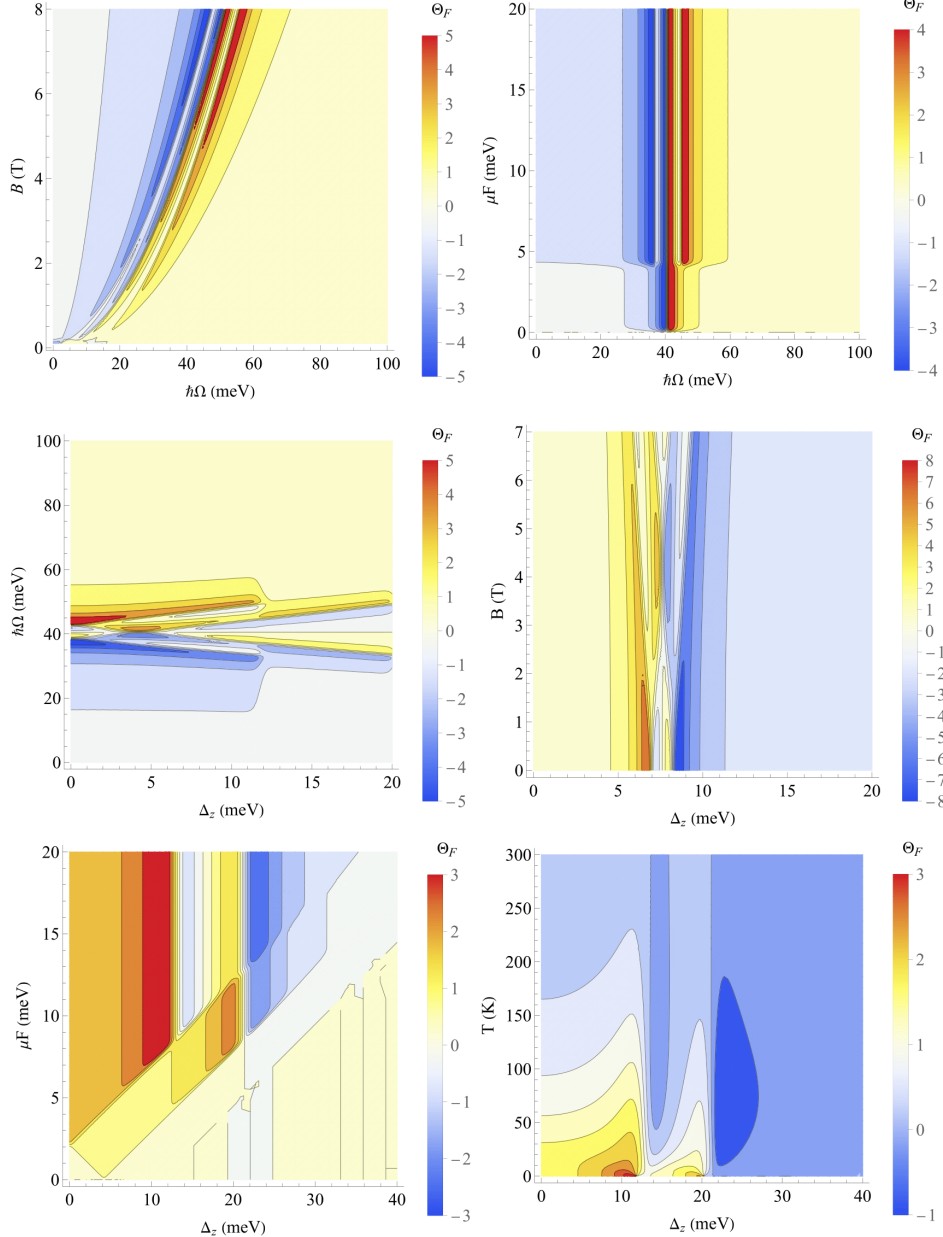

Figure 15: Faraday rotation angle $\Theta_F$ (in degrees) in a silicene monolayer for $\eta = 1$ meV (all) and $\mu_F = 8$ meV, $T = 1$ K, $\hbar\Omega = 50$ meV, $B = 5$ T and $\Delta_z = \Delta_{so}$, when they are not varying.

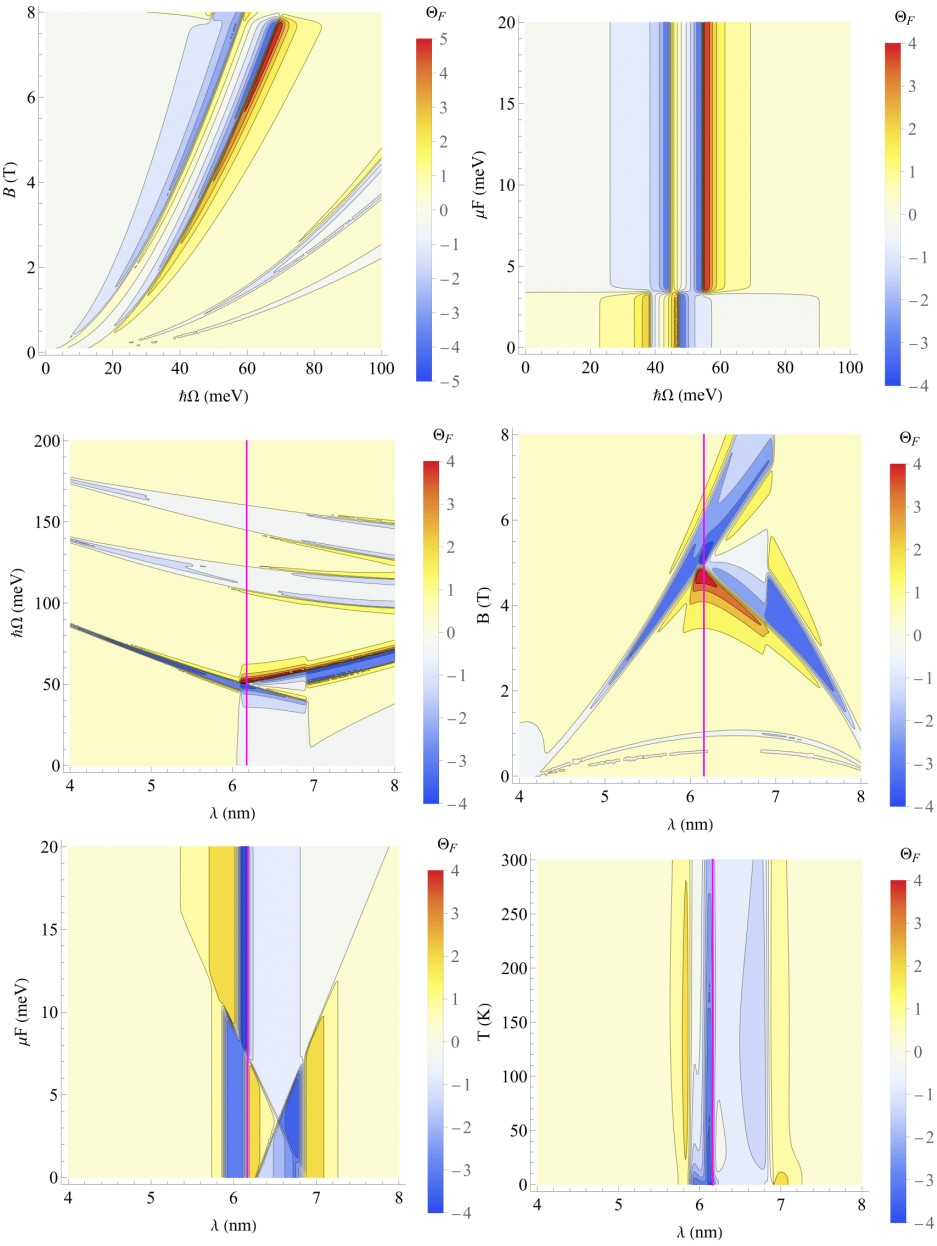

Figure 16: Faraday rotation angle $\Theta_F$ (in degrees) in a bulk HgTe QW for $\eta = 1$ meV (all) and $\mu_F = 8$ meV, $T = 1$ K, $\hbar\Omega = 50$ meV, $B = 5$ T and $\lambda = 6.55$ nm, when they are not varying. The critical point $\lambda_c \simeq 6.17$ nm is marked with a vertical magenta grid line.

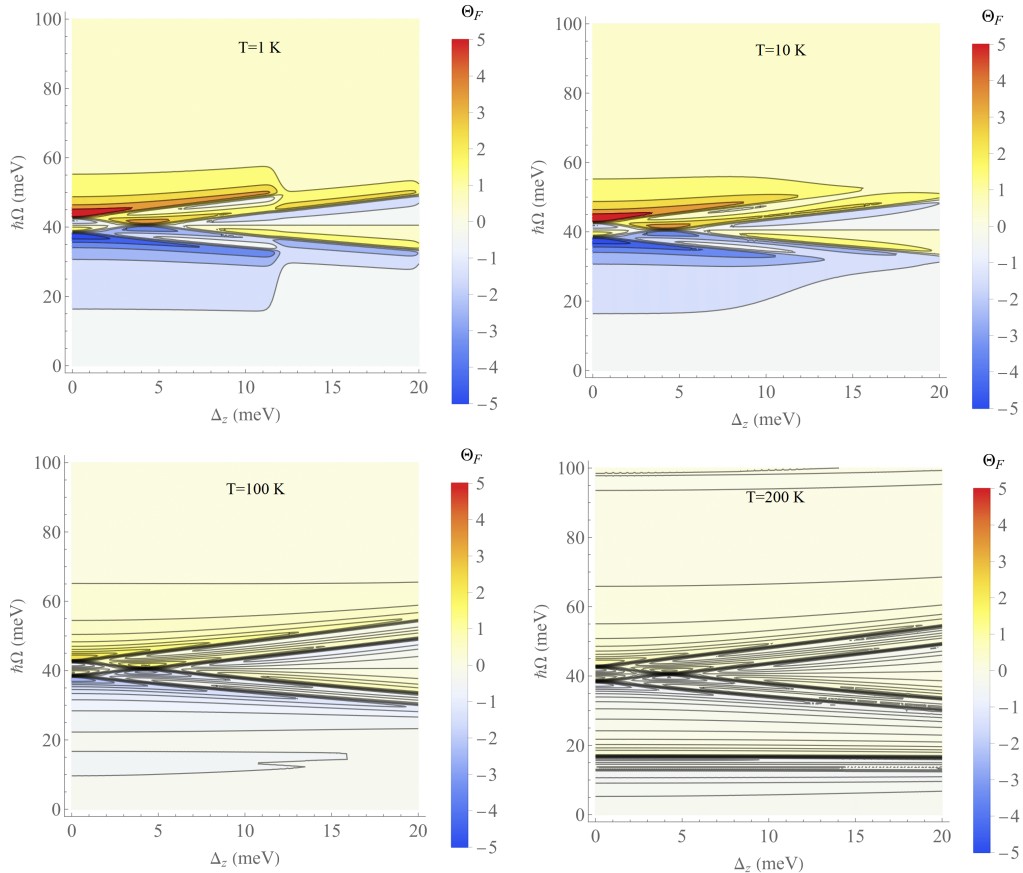

Figure 17: Faraday rotation angle $\Theta_F$ (in degrees) in a silicene monolayer in the parameter space $\{\hbar\Omega, \Delta_z\}$, and for different values of the temperature $T = 1, 10, 100, 200$ K. We set the other parameters as $\eta = 1$ meV, $\mu_F = 8$ meV, and $B = 5$ T.

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
