# Peer review of "Faraday rotation and transmittance as markers of topological phase transitions in 2D materials"

_SciPost Physics, doi:SciPost Phys. 16, 077 (2024)_

## Round 1 · Referee Report · Anonymous · 2023-9-9

Strengths

1 - Well-written
2 - Nice amount of details
3 - Significant research topic

Weaknesses

1 - Needs to improve novelty appealing
2 - Missing citation to recent/important works in the field

Report

Dear editor, below is my report on the manuscript "Faraday rotation and transmittance as markers of topological phase transitions in 2D materials" by M. Calixto, A. Mayorgas, N. A. Cordero, E. Romera, O. Castaños.

In the manuscript, the authors have done an extensive investigation on the magneto-optical properties of different 2D materials. The main claim of the manuscript is to use Faraday rotation and transmittance as possible
markers of topological phase transition (TPT) in the 2D materials. In fact, the reported results show clear signals of TPT in the transmittance (a minimum "peak" value) and in the Faraday rotation (a change of angle sign
with a noticeable peak). Even for the case of non-topological material (phosphorene), a minimum in the transmittance is observed, however associated to the energy gap closing by an external electric field. The manuscript is will written with interesting results and with detailed calculations. However, I have some points that I suggest the authors to revise before further consideration for publication.

Requested changes

1 - The use of magneto-optical effects (linear and also nonlinear optical phenomenon, such as Inverse Faraday Effect) to characterize topological materials have been explored in several previous studies (including Weyl semi-metals), such as: Wu et. al. Science, Vol 354, Issue 6316 pp. 1124-1127 (2016); Ardakani et. al. J. of the Opt. Soc. of Am. B Vol. 38, 9, pp. 2562-2569 (2021); Liang et. al. Opt. Exp. Vol. 28, 17, pp. 24560-24567 (2020); Shao et. al. Nano Lett. 2017, 17, 2, 980–984 (2016); Tokman et. al. Phys. Rev. B 101, 174429 (2020); Shah et. al. Phys. Rev. B 107, 235115 (2023) among many others to cite. In the view of these previous works, it is noticeable that the TPT influences the magneto-optical response, as it is also explored in this present work. Therefore, I suggest the author to give further novel appealing in the Introduction section, so it makes clear why this work would be a "new addition" to the current understanding of TPT signatures on the transmittance and magneto-optical effects.

2 - The results are only for particular direction of magnetic field (perpendicular magnetic field). I believe that an in-plane magnetic field would also be relevant in the context of anisotropy. Therefore, I would suggest the authors to investigate the cases of in-plane for further completeness of the work.

3 - In the beginning of Section II.A, the authors stated that pristine graphene exhibit zero intrinsinc spin-orbit coupling. I agree that it is quite small, specially when compared to the other "heavier" Xenes, with values in the order of \mu eV. However, is not precisely zero, and from the seminal works by Kane and Mele [Phys. Rev. Lett. 95, 226801 (2005) and Phys. Rev. Lett. 95, 226801 (2005)] is also a second-neighbors hopping term, and crucially important to demonstrate the Quantum Spin Hall Effect in graphene.

4 - When showing the results for the phospherene, there is an "important unseen" phenomena which according to the authors does not happen in others 2D materials (just before the section Conclusions). However, there is no proper explanation for this novel phenomena. Therefore, a follow up question is what is the physical origin of this phenomena?

---

## Round 1 · Referee Report · Anonymous · 2023-9-12

Report

In the manuscript by Calixto et al., they present a study of the finite frequency conductivity of 2D Dirac materials under a perpendicular magnetic field. Using a generalized model for the low energy Hamiltonian and adjusting the parameters of the general model, they characterize the topological-band insulator phase transition by studying the conductivity, the transmittance, and the corresponding Faraday rotation of different Dirac materials: Silicene, Germanene, HgTe/CdTe quantum wells, and Phosphorene.

The paper is well-written, the methods seem correct, and the results are sound and relevant. Therefore, I recommend the manuscript publication.

Requested changes

Nevertheless, I suggest the authors:
Establish better which of their results are new contributions and which are a review of previous results.
Include a comparison of the typical plot for Landau levels for all cases studied, perhaps in the supplementary.
Better characterized the effect of temperature in least one of the models.

---

## Round 2 · Referee Report · Anonymous · 2023-11-20

Strengths
1 - Well-written
2 - Scientific sound
3 - Proper citation to the literature
Weaknesses
1 - Results could be further studied for different configurations of external magnetic field
Report
In this revised version of the manuscript, the authors have appropriately responded to my criticism/suggestions. Regardless the fact that the authors have not included results for different directions of magnetic field (which it would be nice to see), I still believe that the results are sound and could be potentially attractive for the readers of SciPost Physics. Therefore, I would recommend the publication of the revised manuscript.

---

## Round 2 · Author Response

Dear Editor,
We carefully address, point by point, the referee’s recommendations and comments in relation to the manuscript REF.: 2305.14923v1 entitled Fara- day rotation and transmittance as markers of topological phase transitions in 2D materials, by M. Calixto, A. Mayorgas, N. Cordero, E. Romera, and O. Castaños, which is submitted to SciPost Physics for publication. See list of changes below.
We thank the referees for their comments and recommendations which helped to improve the article. I hope that the explanations provided and the changes made to the manuscript make it suitable for publication in SciPost Physics.
Looking forward to hearing from you.
Yours sincerely, The Authors

---

## Round 2 · List of Changes

Dear Editor,
We carefully address, point by point, the referee’s recommendations and
comments in relation to the manuscript REF.: 2305.14923v1 entitled Fara-
day rotation and transmittance as markers of topological phase transitions in
2D materials, by M. Calixto, A. Mayorgas, N. Cordero, E. Romera, and O.
Castaños, which is submitted to SciPost Physics for publication.
ANSWER TO REFEREE 1
Point 1) Establish better which of their results are new contributions and
which are a review of previous results.
Answer: A new paragraph has been included at the Introduction (the fifth
one) to establish the new contributions,
“In this paper we perform a comparative . . . the energy gap is closed by an
external electric field."
We have also added a sentence at the end of the abstract for this purpose,
"In the case of non-topological materials as phosphorene, a minimum of the
transmittance is also observed due to the energy gap closing by an external
electric field."
The review of previous results has been extended with a new paragragh in
the Introduction (the third one),
"To determine experimentally the Faraday rotation effect in Dirac materials
it is convenient to...useful for the optical control of magnetization in opto-
electronic devices."
Point 2) Include a comparison of the typical plot for Landau levels for all
cases studied, perhaps in the supplementary.
Answer: We have included, in Sec. I of the Supplementary material, a plot
of the Landau levels vs the external magnetic field B for all three materials
studied.
Point 3) Better characterized the effect of temperature in least one of the
models.
Answer: We have included, in Figure 6 in Sec. V of the Supplementary
material, a series of contour plots of the Faraday angle as a function of the
polarized light frequency Ω and the external electric field ∆z in silicene, where
each subplot has a different value of the temperature.
"In the case of silicene, we have also repeated the contour plot of the param-
eters {ħΩ, ∆z } for different values of the temperature T = 1, 10, 100, 200 K
in Figure 6. The shape of the contour lines is almost the same when varying
the temperature, but oscillation amplitude in the Faraday angle diminish
when increasing T , as the colors of the plots tend to be more flat and yellow
(ΘF ≃ 0 according to the legend)."
ANSWER TO REFEREE 2
Point 1) The use of magneto-optical effects (linear and also nonlinear opti-
cal phenomenon, such as Inverse Faraday Effect) to characterize topological
materials have been explored in several previous studies (including Weyl
semi-metals), such as: Wu et. al. Science, Vol 354, Issue 6316 pp. 1124-1127
(2016); Ardakani et. al. J. of the Opt. Soc. of Am. B Vol. 38, 9, pp.
2562-2569 (2021); Liang et. al. Opt. Exp. Vol. 28, 17, pp. 24560-24567
(2020); Shao et. al. Nano Lett. 2017, 17, 2, 980–984 (2016); Tokman et. al.
Phys. Rev. B 101, 174429 (2020); Shah et. al. Phys. Rev. B 107, 235115
(2023) among many others to cite. In the view of these previous works, it
is noticeable that the TPT influences the magneto-optical response, as it is
also explored in this present work. Therefore, I suggest the author to give
further novel appealing in the Introduction section, so it makes clear why
this work would be a "new addition" to the current understanding of TPT
signatures on the transmittance and magneto-optical effects.
Answer: We thank the referee for these interesting references, which have
been included in a new paragraph of the Introduction (the third one),
"To determine experimentally the Faraday rotation effect in Dirac materials
it is convenient to...useful for the optical control of magnetization in opto-
electronic devices."
We have also included a discussion about the novelty of this work at the fifth
paragraph in the Introduction,
"In this paper we perform a comparative . . . the energy gap is closed by an
external electric field."
Point 2) The results are only for particular direction of magnetic field (per-
pendicular magnetic field). I believe that an in-plane magnetic field would
also be relevant in the context of anisotropy. Therefore, I would suggest the
authors to investigate the cases of in-plane for further completeness of the
work.
Answer: An external in-plane magnetic field is an interesting case to develop
in phosphorene, which has already been studied in Weyl materials in the
reference H. Tan et. al. Phys. Rev. B 103, 214438 (2021). External in-plane
time-periodic laser fields, acting on phosphorene and inducing topological
transitions, are also studied in reference [73] of the new version of the main
article [around the equation (26)]. This problem requires Floquet techniques
(the so called “Floquet topological insulators”, see references [74-76]), which
fall outside the scope of this article. Nevertheless, it deserves to be dealt
with in detail in future work.
Point 3) In the beginning of Section II.A, the authors stated that pristine
graphene exhibit zero intrinsinc spin-orbit coupling. I agree that it is quite
small, specially when compared to the other "heavier" Xenes, with values in
the order of μeV. However, is not precisely zero, and from the seminal works
by Kane and Mele [Phys. Rev. Lett. 95, 226801 (2005) and Phys. Rev.
Lett. 95, 226801 (2005)] is also a secondneighbors hopping term, and cru-
cially important to demonstrate the Quantum Spin Hall Effect in graphene.
Answer: We have removed the sentence "differ from pristine graphene in that
they" in order to avoid the comparison between the spin-orbit coupling of
graphene and other heavier Xenes.
Point 4) When showing the results for the phospherene, there is an "impor-
tant unseen" phenomena which according to the authors does not happen
in others 2D materials (just before the section Conclusions). However, there
is no proper explanation for this novel phenomena. Therefore, a follow up
question is what is the physical origin of this phenomena?
Answer: The unseen phenomena of the phosphorene in Figure 10 is due to
ground state degeneracy at ∆z ≃ −1.53 eV and the parity symmetry of the
Hamiltonian. The main two peaks of the transmittance (energy transitions
E1 odd → E2 even and E0 even → E3 odd ) merge into one for this value of the electric
field, therefore, it is impossible to distinguish both peaks for the given resolu-
tion in η. As a consequence, it is expected that the Faraday angle oscillations
also merge into one, where the peak of the transmittance correspond to a zero
of the Faraday angle in the variable Ω. In the other materials, silicene and
HgTe QWs, only the electronic transitions with the same spin-valley or spin
respectively are allowed, so the oscillations of the Faraday angle vs Ω have
the same direction (they go down-up-down in the increasing Ω direction, see
Figures 5 and 8). However, in phosphorene, due to the Hamiltonian parity
symmetry, only the electronic transitions with different parity are allowed, so
there are Faraday angle oscillations that go down-up-down and others that
go up-down-up. When oscillations with different directions merge into one,
they provoke a peak in the Faraday angle when the transmittance has a peak,
as opposed to the other materials that have a zero value of the Faraday angle
when the transmittance reaches a peak.
The phenomena is not as relevant as the maximum peak of the transmit-
tance in phosphorene, presented in the black line of Figure 10, therefore, we
have removed it from the discussion in the penultimate paragraph of Section
III.C, "However, close to the degeneration point...previously unseen in the
other 2D materials studied in this article."
We thank the referees for their comments and recommendations which
helped to improve the article. I hope that the explanations provided and the
changes made to the manuscript make it suitable for publication in SciPost
Physics.
Looking forward to hearing from you.
Yours sincerely,
The Authors

---

## Editorial Decision

published